# Kernel Stein Tests for Multiple Model Comparison

**Jen Ning Lim**
Max Planck Institute for Intelligent Systems
jlim@tuebingen.mpg.de

**Makoto Yamada**
Kyoto University, RIKEN AIP
makoto.yamada@riken.jp

**Bernhard Schölkopf**
Max Planck Institute for Intelligent Systems
bs@tuebingen.mpg.de

**Wittawat Jitkrittum**
Max Planck Institute for Intelligent Systems
wittawat@tuebingen.mpg.de

## Abstract

We address the problem of non-parametric multiple model comparison: given $l$ candidate models, decide whether each candidate is as good as the best one(s) or worse than it. We propose two statistical tests, each controlling a different notion of decision errors. The first test, building on the post selection inference framework, provably controls the number of best models that are wrongly declared worse (false positive rate). The second test is based on multiple correction, and controls the proportion of the models declared worse but are in fact as good as the best (false discovery rate). We prove that under appropriate conditions the first test can yield a higher true positive rate than the second. Experimental results on toy and real (CelebA, Chicago Crime data) problems show that the two tests have high true positive rates with well-controlled error rates. By contrast, the naive approach of choosing the model with the lowest score without correction leads to more false positives.

## 1  Introduction

Given a sample (a set of i.i.d. observations), and a set of $l$ candidate models $\mathcal{M}$, we address the problem of non-parametric comparison of the relative fit of these candidate models. The comparison is non-parametric in the sense that the class of allowed candidate models is broad (mild assumptions on the models). All the given candidate models may be wrong; that is, the true data generating distribution may not be present in the candidate list. A widely used approach is to pre-select a divergence measure which computes a distance between a model and the sample (e.g., Fréchet Inception Distance (FID, [16]), Kernel Inception Distance [3] or others), and choose the model which gives the lowest estimate of the divergence. An issue with this approach is that multiple equally good models may give roughly the same estimate of the divergence, giving a wrong conclusion of the best model due to noise from the sample (see Table 1 in [17] for an example of a misleading conclusion resulted from direct comparison of two FID estimates).

It was this issue that motivates the development of a non-parametric hypothesis test of relative fit (RelMMD) between two candidate models [4]. The test uses as its test statistic the difference of two estimates of Maximum Mean Discrepancy (MMD, [14]), each measuring the distance between the generated sample from each model and the observed sample. It is known that if the kernel function used is characteristic [27, 11], the population MMD defines a metric on a large class of distributions. As a result, the magnitude of the relative test statistic provides a measure of relative fit, allowing one to decide a (significantly) better model when the statistic is sufficiently large. The key to avoiding the previously mentioned issue of false detection is to appropriately choose the threshold based on the null distribution, i.e., the distribution of the statistic when the two models are equally good. An extension of RelMMD to a linear-time relative test was considered by Jitkrittum et al. [17].

A limitation of the relative tests of RelMMD and others [4, 17] is that they are limited to the comparison of only $l = 2$ candidate models. Indeed, taking the difference is inherently a function of two quantities, and it is unclear how the previous relative tests can be applied when there are $l > 2$ candidate models. We note that relative fit testing is different from goodness-of-fit testing, which aims to decide whether a given model is the true distribution of a set of observations. The latter task may be achieved with the Kernel Stein Discrepancy (KSD) test [6, 23, 13] where, in the continuous case, the model is specified as a probability density function and needs only be known up to the normalizer. A discrete analogue of the KSD test is studied in [32]. When the model is represented by its sample, goodness-of-fit testing reduces to two-sample testing, and may be carried out with the MMD test [14], its incomplete U-statistic variants [33, 31], the ME and SCF tests [7, 18], and related kernel-based tests [8, 10], among others. To reiterate, we stress that in general multiple model comparison differs from multiple goodness-of-fit tests. While the latter may be addressed with $l$ individual goodness-of-fit tests (one for each candidate), the former requires comparing $l$ correlated estimates of the distances between each model and the observed sample. The use of the observed sample in the $l$ estimates is what creates the correlation which must be accounted for.

In the present work, we generalize the relative comparison tests of RelMMD and others [4, 17] to the case of $l > 2$ models. The key idea is to select the "best" model (reference model) that is the closest match to the observed sample, and consider $l$ hypotheses. Each hypothesis tests the relative fit of each candidate model with the reference model, where the reference is chosen to be the model giving the lowest estimate of the pre-chosen divergence measure (MMD or KSD). The total output thus consists of $l$ binary values where 1 (assign positive) indicates that the corresponding model is significantly worse (higher divergence to the sample) than the reference, and 0 indicates no evidence for such claim (indecisive). We assume that the output is always 0 when the reference model is compared to itself. The need for a reference model greatly complicates the formulation of the null hypothesis (i.e., the null hypothesis is random due to the noisy selection of the reference), an issue that is not present in the multiple goodness-of-fit testing.

We propose two non-parametric multiple model comparison tests (Section 3.3) following the previously described scheme. Each test controls a different notion of decision errors. The first test RelPSI builds on the post selection inference framework and provably (Lemma 4.2) controls the number of best models that are wrongly declared worse (FPR, false positive rate). The second test RelMulti is based on multiple correction, and controls the proportion of the models declared worse but are in fact as good as the best (FDR, false discovery rate). In both tests, the underlying divergence measure can be chosen to be either the Maximum Mean Discrepancy (MMD) allowing each model to be represented by its sample, or the Kernel Stein Discrepancy (KSD) allowing the comparison of any models taking the form of unnormalized, differentiable density functions.

As theoretical contribution, the asymptotic null distribution of RelMulti-KSD (RelMulti when the divergence measure is KSD) is provided (Theorem C.1), giving rise to a relative KSD test in the case of $l = 2$ models, as a special case. To our knowledge, this is the first time that a KSD-based relative test for two models is studied. Further, we show (in Theorem 4.1) that the RelPSI test can yield a higher true positive rate (TPR) than the RelMulti test, under appropriate conditions. Experiments (Section 5) on toy and real (CelebA, Chicago Crime data) problems show that the two proposed tests have high true positive rates with well-controlled respective error rates – FPR for RelPSI and FDR for RelMulti. By contrast, the naive approach of choosing the model with the lowest divergence without correction leads to more false positives.

## 2   Background

Hypothesis testing of relative fit between $l = 2$ candidate models, $P_1$ and $P_2$, to the data generating distribution $R$ (unknown) can be performed by comparing the relative magnitudes of a pre-chosen discrepancy measure which computes the distance from each of the two models to the observed sample drawn from $R$. Our proposed methods RelPSI and RelMulti (described in Section 3.3) generalize this formulation based upon selective testing [20], and multiple correction [1], respectively. Underlying these new tests is a base discrepancy measure $D$ for measuring the distance between each candidate model to the observed sample. In this section, we review Maximum Mean Discrepancy (MMD, [14]) and Kernel Stein Discrepancy (KSD, [6, 23]), which will be used as a base discrepancy measure in our proposed tests in Section 3.3.

**Reproducing kernel Hilbert space** Given a positive definite kernel $k : \mathcal{X} \times \mathcal{X} \to \mathbb{R}$, it is known that there exists a feature map $\phi \colon \mathcal{X} \to \mathcal{H}$ and a reproducing kernel Hilbert Space (RKHS) $\mathcal{H}_k = \mathcal{H}$ associated with the kernel $k$ [2]. The kernel $k$ is symmetric and is a reproducing kernel on $\mathcal{H}$ in the sense that $k(x, y) = \langle \phi(x), \phi(y) \rangle_{\mathcal{H}}$ for all $x, y \in \mathcal{X}$ where $\langle \cdot, \cdot \rangle_{\mathcal{H}} = \langle \cdot, \cdot \rangle$ denotes the inner product. It follows from this reproducing property that for any $f \in \mathcal{H}$, $\langle f, \phi(x) \rangle = f(x)$ for all $x \in \mathcal{X}$. We interchangeably write $k(x, \cdot)$ and $\phi(x)$.

**Maximum Mean Discrepancy** Given a distribution $P$ and a positive definite kernel $k$, the mean embedding of $P$, denoted by $\mu_P$, is defined as $\mu_P = \mathbb{E}_{x \sim P}[k(x, \cdot)]$ [26] (exists if $\mathbb{E}_{x \sim P}[\sqrt{k(x, x)}] < \infty$). Given two distributions $P$ and $R$, the Maximum Mean Discrepancy (MMD, [14]) is a pseudometric defined as $\mathrm{MMD}(P, R) := \|\mu_P - \mu_R\|_{\mathcal{H}}$ and $\|f\|_{\mathcal{H}}^2 = \langle f, f \rangle_{\mathcal{H}}$ for any $f \in \mathcal{H}$. If the kernel $k$ is characteristic [27, 11], then MMD defines a metric. An important implication is that $\mathrm{MMD}^2(P, R) = 0 \iff P = R$. Examples of characteristic kernels include the Gaussian and Inverse multiquadric (IMQ) kernels [28, 13]. It was shown in [14] that $\mathrm{MMD}^2$ can be written as $\mathrm{MMD}^2(P, R) = \mathbb{E}_{z, z' \sim P \times R}[h(z, z')]$ where $h(z, z') = k(x, x') + k(y, y') - k(x, y') - k(x', y)$ and $z := (x, y), z' := (x', y')$ are independent copies. This form admits an unbiased estimator $\widehat{\mathrm{MMD}}_u^2 = \frac{1}{n(n-1)} \sum_{i \neq j} h(z_i, z_j)$ where $z_i := (x_i, y_i)$, $\{x_i\}_{i=1}^n \overset{i.i.d.}{\sim} P, \{y_i\}_{i=1}^n \overset{i.i.d.}{\sim} Q$ and is a second-order U-statistic [14]. Gretton et al. [14, Section 6] proposed a linear-time estimator $\widehat{\mathrm{MMD}}_l^2 = \frac{2}{n} \sum_{i=1}^{n/2} h(z_{2i}, z_{2i-1})$ which can be shown to be asymptotically normally distributed both when $P = R$ and $P \neq R$ [14, Corollary 16]. Notice that the MMD can be estimated solely on the basis of two independent samples from the two distributions.

**Kernel Stein Discrepancy** The Kernel Stein Discrepancy (KSD, [23, 6]) is a discrepancy measure between an unnormalized, differentiable density function $p$ and a sample, originally proposed for goodness-of-fit testing. Let $P, R$ be two distributions that have continuously differentiable density functions $p, r$ respectively. Let $\boldsymbol{s}_p(x) := \nabla_x \log p(x)$ (a column vector) be the score function of $p$ defined on its support. Let $k$ be a positive definite kernel with continuous second-order derivatives. Following [23, 19], define $\boldsymbol{\xi}_p(x, \cdot) := \boldsymbol{s}_p(x) k(x, \cdot) + \nabla_x k(x, \cdot)$ which is an element in $\mathcal{H}^d$ that has an inner product defined as $\langle f, g \rangle_{\mathcal{H}^d} = \sum_{i=1}^d \langle f_i, g_i \rangle_{\mathcal{H}}$. The Kernel Stein Discrepancy is defined as $\mathrm{KSD}^2(P, R) := \|\mathbb{E}_{x \sim R} \boldsymbol{\xi}_p(x, \cdot)\|_{\mathcal{H}^d}^2$. Under appropriate boundary conditions on $p$ and conditions on the kernel $k$ [6, 23], it is known that $\mathrm{KSD}^2(P, R) = 0 \iff P = R$. Similarly to the case of MMD, the squared KSD can be written as $\mathrm{KSD}^2(P, R) = \mathbb{E}_{x, x' \sim R}[u_p(x, x')]$ where $u_p(x, x') = \langle \boldsymbol{\xi}_p(x, \cdot), \boldsymbol{\xi}_p(x', \cdot) \rangle_{\mathcal{H}^d} = \boldsymbol{s}_p(x)^\top \boldsymbol{s}_p(x') k(x, x') + \boldsymbol{s}_p(x)^\top \nabla_{x'} k(x, x') + \nabla_x k(x, x')^\top \boldsymbol{s}_p(x') + \mathrm{tr}[\nabla_{x, x'} k(x, x')]$. The KSD has an unbiased estimator $\widehat{\mathrm{KSD}}_u^2(P, R) = \frac{1}{n(n-1)} \sum_{i \neq j} u_p(x_i, x_j)$ where $\{x_i\}_{i=1}^n \overset{i.i.d.}{\sim} R$, which is also a second-order U-statistic. Like the MMD, a linear-time estimator of $\mathrm{KSD}^2$ is given by $\widehat{\mathrm{KSD}}_l^2 = \frac{2}{\lfloor n \rfloor} \sum_{i=1}^{\lfloor n \rfloor / 2} u_p(x_{2i}, x_{2i-1})$. It is known that $\sqrt{n} \widehat{\mathrm{KSD}}_l^2$ is asymptotically normally distributed [23]. In contrast to the MMD estimator, the KSD estimator requires only samples from $R$, and $P$ is represented by its score function $\nabla_x \log p(x)$ which is independent of the normalizing constant. As shown in the previous work, an explicit probability density function is far more representative of the distribution than its sample counterpart [19, 17]. KSD is suitable when the candidate models are given explicitly (i.e., known density functions), whereas MMD is more suitable when the candidate models are implicit and better represented by their samples.

## 3 Proposal: non-parametric multiple model comparison

In this section, we propose two new tests: RelMulti (Section 3.2) and RelPSI (Section 3.3), each controlling a different notion of decision errors.

**Problem** (Multiple Model Comparison). *Suppose we have $l$ models denoted as $\mathcal{M} = \{P_i\}_{i=1}^l$, which we can either: draw a sample (a collection of $n$ i.i.d. realizations) from or have access to their unnormalized log density $\log p(x)$. The goal is to decide whether each candidate $P_i$ is worse than the best one(s) in the candidate list (assign positive), or indecisive (assign zero). The best model is defined to be $P_J$ such that $J \in \arg\min_{j \in \{1, \ldots, l\}} D(P_j, R)$ where $D$ is a base discrepancy measure (see Section 2), and $R$ is the data generating distribution (unknown).*

Throughout this work, we assume that all candidate models $P_1, \ldots, P_l$ and the unknown data generating distribution $R$ have a common support $\mathcal{X} \subseteq \mathbb{R}^d$, and are all distinct. The task can be

seen as a multiple binary decision making task, where a model $P \in \mathcal{M}$ is considered negative if it is as good as the best one, i.e., $D(P, R) = D(P_J, R)$ where $J \in \arg\min_j D(P_j, R)$. The index set of all models which are as good as the best one is denoted by $\mathcal{I}_- := \{i \mid D(P_i, R) = \min_{j=1,\ldots,l} D(P_j, R)\}$. When $|\mathcal{I}_-| > 1$, $J$ is an arbitrary index in $\mathcal{I}_-$. Likewise, a model is considered positive if it is worse than the best model. Formally, the index set of all positive models is denoted by $\mathcal{I}_+ := \{i \mid D(P_i, R) > D(P_J, R)\}$. It follows that $\mathcal{I}_- \cap \mathcal{I}_+ = \emptyset$ and $\mathcal{I}_- \cup \mathcal{I}_+ = \mathcal{I} := \{1, \ldots, l\}$. The problem can be equivalently stated as the task of deciding whether the index for each model belongs to $\mathcal{I}_+$ (assign positive). The total output thus consists of $l$ binary values where 1 (assign positive) indicates that the corresponding model is significantly worse (higher divergence to the sample) than the best, and 0 indicates no evidence for such claim (indecisive). In practice, there are two difficulties: firstly, $R$ can only be observed through a sample $X_n := \{x_i\}_{i=1}^n \overset{i.i.d.}{\sim} R$ so that $D(P_i, R)$ has to be estimated by $\hat{D}(P_i, R)$ computed on the sample; secondly, the index $J$ of the reference model (the best model) is unknown. In our work, we consider the complete, and linear-time U-statistic estimators of MMD or KSD as the discrepancy $\hat{D}$ (see Section 2).

We note that the main assumption on the discrepancy $\hat{D}$ is that $\sqrt{n}(\hat{D}(P_i, R) - \hat{D}(P_j, R)) \overset{d}{\to} \mathcal{N}(\mu, \sigma^2)$ for any $P_i, P_j \in \mathcal{M}$ and $i \neq j$. If this holds, our proposal can be easily amended to accommodate a new discrepancy measure $D$ beyond MMD or KSD. Examples include (but not limited to) the Unnormalized Mean Embedding [7, 17], Finite Set Stein Discrepancy [19, 17], or other estimators such as the block [33] and incomplete estimator [31].

## 3.1 Selecting a reference candidate model

In both proposed tests, the algorithms start by first choosing a model $P_{\hat{j}} \in \mathcal{M}$ as the reference model where $\hat{J} \in \arg\min_{j \in \mathcal{I}} \hat{D}(P_j, R)$ is a random variable. The algorithms then proceed to test the relative fit of each model $P_i$ for $i \neq \hat{J}$ and determine if it is statistically worse than the selected reference $P_{\hat{j}}$. The null and the alternative hypotheses for the $i^{th}$ candidate model can be written as

$$H_{0,i}^{\hat{J}}: D(P_i, R) - D(P_{\hat{j}}, R) \leq 0 \mid P_{\hat{j}} \text{ is selected as the reference,}$$

$$H_{1,i}^{\hat{J}}: D(P_i, R) - D(P_{\hat{j}}, R) > 0 \mid P_{\hat{j}} \text{ is selected as the reference.}$$

These hypotheses are conditional on the selection event (i.e., selecting $\hat{J}$ as the reference index). For each of the $l$ null hypotheses, the test uses as its statistics $\boldsymbol{\eta}^\top \boldsymbol{z} := \sqrt{n}[\hat{D}(P_i, R) - \hat{D}(P_{\hat{j}}, R)]$ where $\boldsymbol{\eta} = [0, \cdots, \underbrace{-1}_{\hat{j}}, \cdots, \underbrace{1}_{i}, \cdots]^\top$ and $\boldsymbol{z} = \sqrt{n}[\hat{D}(P_1, R), \cdots, \hat{D}(P_l, R)]^\top$. The distribution of the test statistic $\boldsymbol{\eta}^\top \boldsymbol{z}$ depends on the choice of estimator for the discrepancy measure $\hat{D}$ which can be $\widehat{\mathrm{MMD}}_u^2$ or $\widehat{\mathrm{KSD}}_u^2$. Define $\boldsymbol{\mu} := [D(P_1, R), \ldots, D(P_l, R)]^\top$, then the hypotheses above can be equivalently expressed as $H_{0,i}^{\hat{J}} : \boldsymbol{\eta}^\top \boldsymbol{\mu} \leq 0 \mid \boldsymbol{A}\boldsymbol{z} \leq \boldsymbol{0}$ vs. $H_{1,i}^{\hat{J}} : \boldsymbol{\eta}^\top \boldsymbol{\mu} > 0 \mid \boldsymbol{A}\boldsymbol{z} \leq \boldsymbol{0}$, where we note that $\boldsymbol{\eta}$ depends on $i$, $\boldsymbol{A} \in \{-1, 0, 1\}^{(l-1) \times l}$, $\boldsymbol{A}_{s,:} = [0, \ldots, \underbrace{1}_{\hat{j}}, \cdots, \underbrace{-1}_{s}, \cdots, 0]$ for all $s \in \{1, \ldots, l\} \backslash \{\hat{J}\}$ and $\boldsymbol{A}_{s,:}$ denote the $s^{th}$ row of $\boldsymbol{A}$. This equivalence was exploited in the multiple goodness-of-fit testing by Yamada et al. [31]. The condition $\boldsymbol{A}\boldsymbol{z} \leq \boldsymbol{0}$ represents the fact that $P_{\hat{j}}$ is selected as the reference model, and expresses $\hat{D}(P_{\hat{j}}, R) \leq \hat{D}(P_s, R)$ for all $s \in \{1, \ldots, l\} \backslash \{\hat{J}\}$.

## 3.2 RelMulti: for controlling false discovery rate (FDR)

Unlike traditional hypothesis testing, the null hypotheses here are conditional on the selection event, making the null distribution non-trivial to derive [21, 22]. Specifically, the sample used to form the selection event (i.e., establishing the reference model) is the same sample used for testing the hypothesis, creating a dependency. Our first approach of RelMulti is to divide the sample into two independent sets, where the first is used to choose $P_{\hat{j}}$ and the latter for performing the test(s). This approach simplifies the null distribution since the sample used to form the selection event and the test sample are now independent. That is, $H_{0,i}^{\hat{J}} : \boldsymbol{\eta}^\top \boldsymbol{\mu} \leq 0 \mid \boldsymbol{A}\boldsymbol{z} \leq \boldsymbol{0}$ simplifies to $H_{0,i}^{\hat{J}} : \boldsymbol{\eta}^\top \boldsymbol{\mu} \leq 0$ due to independence. In this case, the distribution of the test statistic (for $\widehat{\mathrm{MMD}}_u^2$ and $\widehat{\mathrm{KSD}}_u^2$) after

selection is the same as its unconditional null distribution. Under our assumption that all distributions are distinct, the test statistic is asymptotically normally distributed [14, 23, 6].

For the complete U-statistic estimator of Maximum Mean Discrepancy ($\widehat{\text{MMD}}_u^2$), Bounliphone et al. [4] showed that, under mild assumptions, $z$ is jointly asymptotically normal, where the covariance matrix is known in closed form. However, for $\widehat{\text{KSD}}_u^2$, only the marginal variance is known [6, 23] and not its cross covariances, which are required for characterizing the null distributions of our test (see Algorithm 2 in the appendix for the full algorithm of RelMulti). We present the asymptotic multivariate characterization of $\widehat{\text{KSD}}_u^2$ in Theorem C.1.

Given a desired significance level $\alpha \in (0, 1)$, the rejection threshold is chosen to be the $(1 - \alpha)$-quantile of the distribution $\mathcal{N}(0, \hat{\sigma}^2)$ where $\hat{\sigma}^2$ is the plug-in estimator of the asymptotic variance $\sigma^2$ of our test statistic (see [4, Section 3] for MMD and Section D for KSD). With this choice, the false rejection rate for each of the $l - 1$ hypotheses is upper bounded by $\alpha$ (asymptotically). However, to control the false discovery rate for the $l - 1$ tests it is necessary to further correct with multiple testing adjustments. We use the Benjamini–Yekutieli procedure [1] to adjust $\alpha$. We note that when testing $H_{0,\hat{J}}^{\hat{J}}$, the result is always 0 (fail to reject) by default. When $l > 2$, following the result of [1] the asymptotic false discovery rate (FDR) of RelMulti is provably no larger than $\alpha$. The FDR in our case is the fraction of the models declared worse that are in fact as good as the (true) reference model. For $l = 2$, no correction is required as only one test is performed.

### 3.3 RelPSI: for controlling false positive rate (FPR)

A caveat of the data splitting used in RelMulti is the loss of true positive rate since a portion of sample for testing is used for forming the selection. When the selection is wrong, i.e., $\hat{J} \in \mathcal{I}_+$, the test will yield a lower true positive rate. It is possible to alleviate this issue by using the full sample for selection and testing, which is the approach taken by our second proposed test RelPSI. This approach requires us to know the null distribution of the conditional null hypotheses (see Section 3.1), which can be derived based on Theorem 3.1.

**Theorem 3.1** (Polyhedral Lemma [20])**.** *Suppose that $z \sim \mathcal{N}(\mu, \Sigma)$ and the selection event is affine, i.e., $Az \leq b$ for some matrix $A$ and $b$, then for any $\eta$, we have*

$$\eta^\top z \mid Az \leq b \sim \mathcal{TN}(\eta^\top \mu, \ \eta^\top \Sigma \eta, \ \mathcal{V}^-(z), \ \mathcal{V}^+(z)),$$

*where $\mathcal{TN}(\mu, \sigma^2, a, b)$ is a truncated normal distribution with mean $\mu$ and variance $\sigma^2$ truncated at $[a, b]$. Let $\alpha = \frac{A\Sigma\eta}{\eta^\top \Sigma \eta}$. The truncated points are given by: $\mathcal{V}^-(z) = \max_{j:\alpha_j < 0} \frac{b_j - Az_j}{\alpha_j} + \eta^\top z$, and $\mathcal{V}^+(z) = \min_{j:\alpha_j > 0} \frac{b_j - Az_j}{\alpha_j} + \eta^\top z$.*

This lemma assumes two parameters are known: $\mu$ and $\Sigma$. Fortunately, we do not need to estimate $\mu$ and can set $\eta^\top \mu = 0$. To see this note that threshold is given by $(1 - \alpha)$-quantile of a truncated normal which is $t_\alpha := \eta^\top \mu + \sigma \Phi^{-1} \big( (1 - \alpha) \Phi \big( \frac{\mathcal{V}^+ - \eta^\top \mu}{\sigma} \big) + \alpha \Phi \big( \frac{\mathcal{V}^- - \eta^\top \mu}{\sigma} \big) \big)$ where $\sigma^2 = \eta^\top \Sigma \eta$. If our test statistic $\eta^\top z$ exceeds the threshold, we reject the null hypothesis $H_{0,i}^{\hat{J}}$. This choice of the rejection threshold will control the *selective type-I error* $\mathbb{P}(\eta^\top z > t_\alpha \mid H_{0,i}^{\hat{J}}$ is true, $P_{\hat{J}}$ is selected) to be no larger than $\alpha$. However $\mu$ is not known, the threshold can be adjusted by setting $\eta^\top \mu = 0$ and can be seen as a more conservative threshold. A similar adjustment procedure is used in Bounliphone et al. [4] and Jitkrittum et al. [17] for Gaussian distributed test statistics. And since $\Sigma$ is also unknown, we replace $\Sigma$ with a consistent plug-in estimator $\hat{\Sigma}$ given by Bounliphone et al. [4, Theorem 2] for $\widehat{\text{MMD}}_u^2$ and Theorem C.1 for $\widehat{\text{KSD}}_u^2$. Specifically, we have as the threshold $\hat{t}_\alpha := \hat{\sigma} \Phi^{-1} \big( (1 - \alpha) \Phi \big( \frac{\mathcal{V}^+}{\hat{\sigma}} \big) + \alpha \Phi \big( \frac{\mathcal{V}^-}{\hat{\sigma}} \big) \big)$ where $\hat{\sigma}^2 = \eta^\top \hat{\Sigma} \eta$ (see Algorithm 1 in the appendix for the full algorithm of RelPSI).

Our choice of $\eta$ depends on the realization of $\hat{J}$, but $\eta$ can be fixed such that the test we perform is independent of our observation of $\hat{J}$ (see Experiment 1). For a fixed $\eta$, the concept of power, i.e., $\mathbb{P}(\eta^\top z > \hat{t}_\alpha)$ when $\eta^\top \mu > 0$, is meaningful; and we show in Theorem 3.2 that our test is consistent using MMD. However, when $\eta$ is random (i.e., dependent on $\hat{J}$) the notion of test power is less appropriate, and we use true positive rate and false positive rate to measure the performance (see Section 4).

**Theorem 3.2** (Consistency of RelPSI-MMD). *Given two models $P_1$, $P_2$ and a data distribution $R$ (which are all distinct). Let $\hat{\Sigma}$ be a consistent estimate of the covariance matrix defined in Theorem C.2. and $\boldsymbol{\eta}$ be defined such that $\boldsymbol{\eta}^\top \boldsymbol{z} = \sqrt{n}[\widehat{\mathrm{MMD}}_u^2(P_2, R) - \widehat{\mathrm{MMD}}_u^2(P_1, R)]$. Suppose that the threshold $\hat{t}_\alpha$ is the $(1 - \alpha)$-quantile of $\mathcal{TN}(\mathbf{0}, \boldsymbol{\eta}^\top \hat{\Sigma} \boldsymbol{\eta}, \mathcal{V}^-, \mathcal{V}^+)$ where $\mathcal{V}^+$ and $\mathcal{V}^-$ are defined in Theorem 3.1. Under $H_0 : \boldsymbol{\eta}^\top \boldsymbol{\mu} \leq 0 \,|\, P_{\hat{j}}$ is selected, the asymptotic type-I error is bounded above by $\alpha$. Under $H_1 : \boldsymbol{\eta}^\top \boldsymbol{\mu} > 0 \,|\, P_{\hat{j}}$ is selected, we have $\mathbb{P}(\boldsymbol{\eta}^\top \boldsymbol{z} > \hat{t}_\alpha) \to 1$ as $n \to \infty$.*

A proof for Theorem 3.2 can be found in Section G in the appendix. A similar result holds for RelPSI-KSD (see Appendix G.1) whose proof follows closely the proof of Theorem 3.2 and is omitted.

# 4 Performance analysis

Post selection inference (PSI) incurs its loss of power from conditioning on the selection event [9, Section 2.5]. Therefore, in the fixed hypothesis (not conditional) setting of $l = 2$ models, it is unsurprising that the empirical power of RelMMD and RelKSD is higher than its PSI counterparts (see Experiment 1). However, when $l = 2$, and conditional hypotheses are considered, it is unclear which approach is desirable. Since both PSI (as in RelPSI) and data-splitting (as in RelMulti) approaches for model comparison have tractable null distributions, we study the performance of our proposals for the case when the hypothesis is dependent on the data.

We measure the performance of RelPSI and RelMulti by *true positive rate* (TPR) and *false positive rate* (FPR) in the setting of $l = 2$ candidate models. These are popular metrics when reporting the performance of selective inference approaches [29, 31, 9]. TPR is the expected proportion of models worse than the best that are correctly reported as such. FPR is the expected proportion of models as good as the best that are wrongly reported as worse. It is desirable for TPR to be high and FPR to be low. We defer the formal definitions to Section A (appendix); when we estimate TPR and FPR, we denote it as $\widehat{\mathrm{TPR}}$ and $\widehat{\mathrm{FPR}}$ respectively. In the following theorem, we show that the TPR of RelPSI is higher than the TPR of RelMulti.

**Theorem 4.1** (TPR of RelPSI and RelMulti). *Let $P_1, P_2$ be two candidate models, and $R$ be a data generating distribution. Assume that $P_1, P_2$ and $R$ are distinct. Given $\alpha \in [0, \frac{1}{2}]$ and split proportion $\rho \in (0, 1)$ for RelMulti so that $(1 - \rho)n$ samples are used for selecting $P_{\hat{j}}$ and $\rho n$ samples for testing, for all $n \gg N = \left( \frac{\sigma \Phi^{-1}(1 - \frac{\alpha}{2})}{\mu(1 - \sqrt{\rho})} \right)^2$, we have $\mathrm{TPR}_{RelPSI} \gtrsim \mathrm{TPR}_{RelMulti}$.*

The proof is provided in the Section F.6. This result holds for both MMD and KSD. Additionally, in the following result we show that both approaches bound FPR by $\alpha$. Thus, RelPSI controls FPR regardless of the choice of discrepancy measure and number of candidate models.

**Lemma 4.2** (FPR Control). *Define the selective type-I error for the $i^{th}$ model to be $s(i, \hat{J}) := \mathbb{P}(reject\ H_{0,i}^{\hat{J}} \,|\, H_{0,i}^{\hat{J}}\ is\ true, P_{\hat{j}}\ is\ selected)$. If $s(i, \hat{J}) \leq \alpha$ for all $i, \hat{J} \in \{1, \dots, l\}$, then $\mathrm{FPR} \leq \alpha$.*

The proof can be found in Section A. For both RelPSI and RelMulti, the test threshold is chosen to control the selective type-I error. Therefore, both control FPR to be no larger than $\alpha$. In RelPSI, we explicitly control this quantity by characterizing the distribution of statistic under the conditional null.

**Remark.** *The selection of the best model is a noisy process, and we can pick a model that is worse than the actual best, i.e., $\hat{J} \notin \arg\min_j D(P_j, R)$. An incorrect selection results in a higher portion of true conditional null hypotheses. So, the true positive rate of the test will be lowered. However, the false rejection is still controlled at level $\alpha$.*

# 5 Experiments

In this section, we demonstrate our proposed method for both toy problems and real world datasets. Our first experiment is a baseline comparison of our proposed method RelPSI to RelMMD [4] and RelKSD (see Appendix D). In this experiment, we consider a fixed hypothesis of model comparison for two candidate models (RelMulti is not applicable here). This is the original setting that RelMMD was proposed for. In the second experiment, we consider a set of mixture models for smiling and non-smiling images of CelebA [24] where each model has its own unique generating proportions

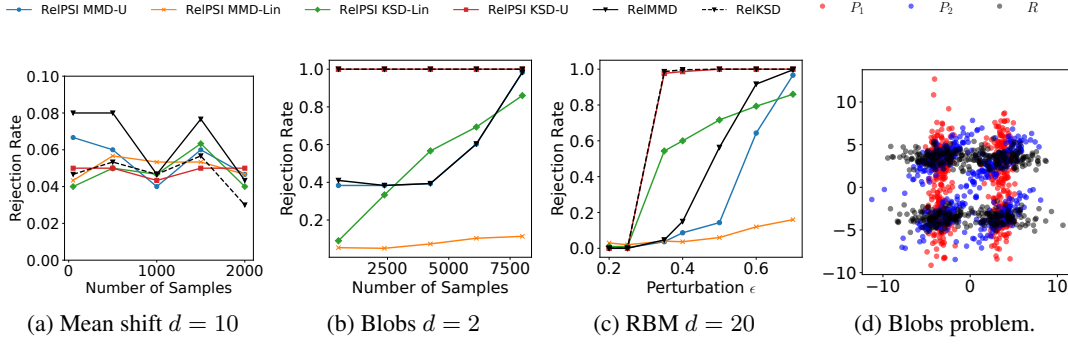

Figure 1: Rejection rates (estimated from 300 trials) for the six tests with $\alpha = 0.05$ is shown. "MMD-U" refers to the usage of the complete U-statistic for MMD which is $\widehat{\mathrm{MMD}}_u^2$, "MMD-Lin" refers to the linear time estimator $\widehat{\mathrm{MMD}}_l^2$ and similarly for KSD Complete and KSD Linear (defined in Section 2).

from the real data set or images from trained GANs. For our final experiment, we examine density estimation models trained on the Chicago crime dataset considered by Jitkrittum et al. [19]. In this experiment, each model has a score function which allows us to apply both RelPSI and RelMulti with KSD. In the last two experiments on real data, there is no ground truth for which candidate model is the best; so estimating TPR, FDR and FPR is infeasible. Instead, the experiments are designed to have a strong indication of the ground truth with support from another metric. More synthetic experiments are shown in Appendix H to verify our theoretical results.

The kernel parameters used in the discrepancy between each model $P_i$ and the data distribution $R$ are the same to ensure the comparison between the discrepancies are meaningful. If the median heuristic is used, the bandwidth parameter is the empirical median of all the pairwise L2 distances between the given samples. For MMD, samples from the data distribution $R$ and all the model samples $\mathcal{M}$ are used to calculate the median heuristic. Whereas for KSD, only the samples from $R$ are used. Code for reproducing the results can be found online.[1] We note that to account for sample variability, our experiments are averaged over at least 100 trials with new samples (from a different seed) redrawn for each trial.

**1. A comparison of** RelMMD, RelKSD, RelPSI-KSD **and** RelPSI-MMD ($l = 2$): The aim of this experiment is to investigate the behaviour of the proposed tests with RelMMD and RelKSD as baseline comparisons and empirically demonstrate that RelPSI-MMD and RelPSI-KSD possess desirable properties such as level-$\alpha$ and comparable test power. Since RelMMD and RelKSD have no concept of selection, in order for the results to be comparable we fixed null hypothesis to be $H_0 : D(P_1, R) \leq D(P_2, R)$ which is possible for RelPSI by fixing $\boldsymbol{\eta}^\top = [-1, 1]$. In this experiment, we consider the following problems:

1. *Mean shift*: The two candidate models are isotropic Gaussians on $\mathbb{R}^{10}$ with varying mean: $P_1 = \mathcal{N}([0.5, 0, \cdots, 0], \boldsymbol{I})$ and $P_2 = \mathcal{N}([-0.5, 0, \cdots, 0], \boldsymbol{I})$. Our reference distribution is $R = \mathcal{N}(\boldsymbol{0}, \boldsymbol{I})$. In this case, $H_0$ is true.

2. *Blobs*: This problem was studied by various authors [7, 15, 17]. Each distribution is a mixture of Gaussians with a similar structure on a global scale but different locally by rotation. Samples from this distribution is shown in Figure 1d. In this case, the $H_1$ is true.

3. *Restricted Boltzmann Machine (RBM)*: This problem was studied by [23, 19, 17]. Each distribution is given by a Gaussian Restricted Boltzmann Machine (RBM) with a density $p(\boldsymbol{y}) = \sum_{\boldsymbol{x}} p'(\boldsymbol{y}, \boldsymbol{x})$ and $p'(\boldsymbol{y}, \boldsymbol{x}) = \frac{1}{Z} \exp(\boldsymbol{y}^\top \boldsymbol{B} \boldsymbol{x} + \boldsymbol{b}^\top \boldsymbol{y} + \boldsymbol{c}^\top \boldsymbol{x} - \frac{1}{2}||\boldsymbol{y}||^2)$ where $\boldsymbol{x}$ are the latent variables and model parameters are $\boldsymbol{B}, \boldsymbol{b}, \boldsymbol{c}$. The model will share the same parameters $\boldsymbol{b}$ and $\boldsymbol{c}$ (which are drawn from a standard normal) with the reference distribution but the matrix $\boldsymbol{B}$ (sampled uniformly from $\{-1, 1\}$) will be perturbed with $\boldsymbol{B}^{p_2} = \boldsymbol{B} + 0.3\delta$ and $\boldsymbol{B}^{p_1} = \boldsymbol{B} + \epsilon\delta$ where $\epsilon$ varies between 0 and 1. It measures the sensitivity of the test [19]

Table 1: A comparison of our proposed method with FID. The underlying distribution are samples forming a mixture of smiling (S) or non-smiling (N) faces which can be either generated (G) or real (R). "Rej." denotes the rate of rejection of the model indicating that it is significantly worse than the best model. "Sel." is the rate at which the model is selected (the one with the minimum discrepancy score). Average FID scores are also reported. These results are averaged over 100 trials.

| Model | Mix | | RelPSI-MMD | | RelMulti-MMD | | FID | |
|---|---|---|---|---|---|---|---|---|
| | S | N | Rej. | Sel. | Rej. | Sel. | Aver. | Sel. |
| 1 | 0.50 (G) | 0.50 (G) | 0.99 | 0.0 | 1.0 | 0.0 | $27.86 \pm 0.49$ | 0 |
| 2 | 0.60 (R) | 0.40 (R) | 0.39 | 0.02 | 0.18 | 0.08 | $16.01 \pm 0.19$ | 0.39 |
| 3 | 0.40 (R) | 0.60 (R) | 0.28 | 0.03 | 0.19 | 0.10 | $16.29 \pm 0.20$ | 0.03 |
| 4 | 0.51 (R) | 0.49 (R) | 0.02 | 0.52 | 0.03 | 0.37 | $16.03 \pm 0.18$ | 0.27 |
| 5 | 0.52 (R) | 0.48 (R) | 0.06 | 0.43 | 0.0 | 0.45 | $16.01 \pm 0.17$ | 0.31 |
| Truth | 0.5 (R) | 0.5 (R) | - | - | - | - | - | - |

since perturbing only one entry can create a difference that is hard to detect. Furthermore, We fix $n = 1000$, $d_x = 5$, $d_y = 20$.

Our proposal and baselines are all non-parametric kernel based test. For a fair comparison, all the tests use the same Gaussian kernel with its bandwidth chosen by the median heuristic. In Figure 1, it shows the rejection rates for all tests. As expected, the tests based on KSD have higher power than MMD due to having access to the density function. Additionally, linear time estimators perform worse than their complete counterpart.

In Figure 1a, when $H_0$ is true, then the false rejection rate (type-I error) is controlled around level $\alpha$ for all tests. In Figure 1b, the poor performance of MMD-based tests in blobs experiments is caused by an unsuitable choice of bandwidth. The median heuristic cannot capture the small-scale differences [15, 17]. Even though KSD-based tests utilize the same heuristic, equipped with the density function a mismatch in the distribution shape can be detected. Interestingly, in all our experiments, the RelPSI variants perform comparatively to their cousins, Rel-MMD and Rel-KSD but as expected, the power is lowered due to the loss of information from our conditioning [9]. These two problems show the behaviour of the tests when the number of samples $n$ increases.

In Figure 1c, this shows the behaviour of the tests when the difference between the candidate models increases (one model gets closer to the reference distribution). When $\epsilon < 0.3$, the null case is true and the tests exhibit a low rejection rate. However, when $\epsilon > 0.3$ then the alternative is true. Tests utilizing KSD can detect this change quickly which indicated by the sharp increase in the rejection rate when $\epsilon = 0.3$. However, MMD-based tests are unable to detect the differences at that point. As the amount of perturbation increases, this changes and MMD tests begin to identify with significance that the alternative is true. Here we see that RelPSI-MMD has visibly lowered rejection rate indicating the cost of power for conditioning, whilst for RelPSI-KSD and RelKSD both have similar power.

**2. Image Comparison** ($l = 5$): In this experiment, we apply our proposed test RelPSI-MMD and RelMulti-MMD for comparing between five image generating candidate models. We consider the CelebA dataset [24] which for each sample is an image of a celebrity labelled with 40 annotated features. As our reference distribution and candidate models, we use a mixture of smiling and non-smiling faces of varying proportions (Shown in Table 1) where the model can generate images from a GAN or from the real dataset. For generated images, we use the GANs of [17, Appendix B]. In each trial, $n = 2000$ samples are used. We partition the dataset such that the reference distribution draws distinct independent samples, and each model samples independently of the remainder of the pool. All algorithms receive the same model samples. The kernel used is the Inverse Multiquadric (IMQ) on 2048 features extracted by the Inception-v3 network at the pool layer [30]. Additionally, we use 50:50 split for RelMulti-MMD. Our baseline is the procedure of choosing the lowest Fréchet Inception Distance (FID) [16]. We note the authors did not propose a statistical test with FID. Table 1 summaries the results from the experiment.

In Table 1, we report the model-wise rejection rate (a high rejection indicts a poor candidate relatively speaking) and the model selection rate (which indicates the rate that the model has the smallest discrepancy from the given samples). The table illustrates several interesting points. First, even

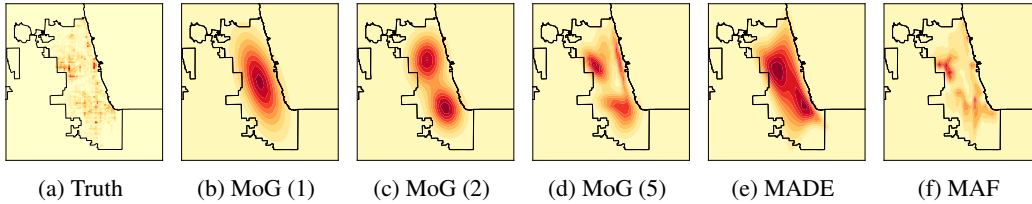

| (a) Truth | (b) MoG (1) | (c) MoG (2) | (d) MoG (5) | (e) MADE | (f) MAF |

Figure 2: The density plots of the trained models on the Chicago Crime dataset.

though Model 1 shares the same portions as the true reference models, the quality of the generated images is a poor match to the reference images and thus is frequently rejected. A considerably higher FID score (than the rest) also supports this claim. Secondly, in this experiment, MMD is a good estimator of the best model for both RelPSI and RelMulti (with splitting exhibiting higher variance) but the minimum FID score selects the incorrect model $73\%$ of the time. The additional testing indicate that Model 4 or Model 5 could be the best as they were rarely deemed worse than the best which is unsurprising given that their mixing proportions are closest to the true distribution. The low rejection for Model 4 is expected given that they differ by only $40$ samples. Model 2 and 3 have respectable model-wise rejections to indicate their position as worse than the best. Overall, both RelPSI and RelMulti perform well and shows that the additional testing phase yields more information than the approach of picking the minimum of a score function (especially for FID).

**3. Density Comparison** $(l = 5)$: In our final experiment, we demonstrate RelPSI-KSD and RelMulti-KSD on the Chicago data-set considered in Jitkrittum et al. [19] which consists of 11957 data points. We split the data-set into disjoint sets such that 7000 samples are used for training and the remainder for testing. For our candidate models, we trained a Mixture of Gaussians (MoG) with expectation maximization with $C$ components where $C \in \{1, 2, 5\}$, Masked Auto-encoder for Density Estimation (MADE) [12] and a Masked Auto-regressive Flow (MAF) [25]. MAF with 1 autoregressive layer with a standard normal as the base distribution (or equivalently MADE) and MAF model has 5 autoregressive layers with a base distribution of a MoG (5). Each autoregressive layer is a feed-forward network with 512 hidden units. Both invertible models are trained with maximum likelihood with a small amount of $\ell_2$ penalty on the weights. In each trial, we sample $n = 2000$ points independently of the test set. The resultant density shown in Figure 2 and the reference distribution in Figure 2a. We compare our result with the negative log-likelihood (NLL). Here we use the IMQ kernel.

The results are shown in Table 2. If performance is measured by a higher model-wise rejection rates, for this experiment RelPSI-KSD performs better than RelMulti-KSD. RelPSI-KSD suggests that MoG (1), MoG (2) and MADE are worse than the best but is unsure about MoG (5) and MAF. Whilst the only significant rejection of RelMulti-KSD is MoG (1). These findings with RelPSI-KSD can be further endorsed by inspecting the density (see Figure 2). It is clear that MoG (1), MoG (2) and MADE are too simple. But between MADE and MAF (5), it is unclear which is a better fit. Negative Log Likelihood (NLL) consistently suggest that MAF is the best which corroborates with our findings that MAF is one of the top models. The preference of MAF for NLL is due to log likelihood not penalizing the complexity of the model (MAF is the most complex with the highest number of parameters).

|  | RelPSI-KSD | | RelMul-KSD | | NLL | |
| --- | --- | --- | --- | --- | --- | --- |
| Model | Rej. | Sel. | Rej. | Sel. | Aver. | Sel. |
| MoG (1) | 0.42 | 0. | 0.22 | 0 | 2.64 | 0 |
| MoG (2) | 0.28 | 0.01 | 0.07 | 0.08 | 2.55 | 0 |
| MoG (5) | 0.02 | 0.62 | 0 | 0.38 | 2.38 | 0 |
| MADE | 0.26 | 0.01 | 0.04 | 0.03 | 2.53 | 0 |
| MAF (5) | 0 | 0.36 | 0 | 0.51 | 2.25 | 1. |

Table 2: Relative testing on unconditional density estimation models. The model-wise rejection rates, selection rates and average negative log likelihood (NLL) scores are reported. These results are averaged over 100 trials.

**Acknowledgments**

M.Y. was supported by the JST PRESTO program JPMJPR165A and partly supported by MEXT KAKENHI 16H06299 and the RIKEN engineering network funding.

## Footnotes

[1] https://github.com/jenninglim/model-comparison-test

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
