[Supplementary Material]

# Kernel Stein Tests for Multiple Model Comparison

## Supplementary

## A    Definitions and FPR proof

In this section, we define TPR and FPR, and prove that our proposals control FPR.

Recall the definitions of $\mathcal{I}_-$ and $\mathcal{I}_+$ (see Section 3.3). $\mathcal{I}_-$ is the set of models that are *not* worse than $P_J$ (the true best model). $\mathcal{I}_+$ is the set of models that are worse than $P_J$. We say that an algorithm decides that a model $P_i$ is positive if it decides that $P_i$ is worse than $P_J$. We define true positive rate (TPR) and false positive rate (FPR) to be

$$\text{FPR} = \frac{1}{|\mathcal{I}_-|}\mathbb{E}[|\{i \in \mathcal{I}_- \ : \ \text{the algorithm decides that } P_i \text{ is positive}\}|],$$

$$\text{TPR} = \frac{1}{|\mathcal{I}_+|}\mathbb{E}[|\{i \in \mathcal{I}_+ \ : \ \text{the algorithm decides that } P_i \text{ is positive}\}|].$$

Both TPR and FPR can be estimated by averaging the TPR and FPR with multiple independent trials (as was done in Experiment H). We call this quantity the empirical TPR and FPR, denoted as $\widehat{\text{TPR}}$ and $\widehat{\text{FPR}}$ respectively.

The following lemma shows that our proposals controls FPR.

**Lemma A.1** (FPR Control). *Define the selective type-I error for the $i^{th}$ model to be $s(i, \hat{J}) := \mathbb{P}(\text{reject } H_{0,i}^{\hat{J}} \mid H_{0,i}^{\hat{J}} \text{ is true}, P_{\hat{j}} \text{ is selected})$. If $s(i, \hat{J}) \leq \alpha$ for all $i, \hat{J} \in \{1, \ldots, l\}$, then $\text{FPR} \leq \alpha$.*

*Proof.* From law of total expectation, we have

$$\text{FPR} = \frac{1}{|\mathcal{I}_-|}\mathbb{E}[|\{i \in \mathcal{I}_- \ : \ \text{the algorithm decides that } i \text{ is positive}\}|]$$

$$= \frac{1}{|\mathcal{I}_-|}\mathbb{E}[\mathbb{E}[|\{i \in \mathcal{I}_- \ : \ \text{the algorithm decides that } i \text{ is positive}\}| \mid P_{\hat{j}} \text{ is selected}]]$$

$$= \frac{1}{|\mathcal{I}_-|}\mathbb{E}[\mathbb{E}[\sum_{i \in \mathcal{I}_-} \mathbb{I}(\text{The algorithm decides that } i \text{ is positive}) \mid P_{\hat{j}} \text{ is selected}]]$$

$$= \frac{1}{|\mathcal{I}_-|}\mathbb{E}[\sum_{i \in \mathcal{I}_-} \mathbb{P}(\text{The algorithm decides that } i \text{ is positive} \mid P_{\hat{j}} \text{ is selected})]$$

$$= \frac{1}{|\mathcal{I}_-|}\mathbb{E}[\sum_{i \in \mathcal{I}_-} \mathbb{P}(\sqrt{n}[\hat{D}(P_i, R) - \hat{D}(P_{\hat{j}}, R)] > \hat{t}_\alpha \mid P_{\hat{j}} \text{ is selected})]$$

$$\leq \frac{1}{|\mathcal{I}_-|}\mathbb{E}[\sum_{i \in \mathcal{I}_-} \alpha]$$

$$= \frac{1}{|\mathcal{I}_-|}|\mathcal{I}_-|\alpha$$

$$= \alpha,$$

where $\mathbb{I}$ is the indicator function.                                                                       $\square$

## B    Algorithms

Algorithms for RelPSI (see Algorithm 1) and RelMulti (see Algorithm 2) proposed in Section 3 are provided in this section.

In algorithm 2, `FDRCorrection`($\boldsymbol{p}, \alpha$) takes a list of $p$-values $\boldsymbol{p}$ and returns a list of rejections for each element of $\boldsymbol{p}$ such that the false discovery rate is controlled at $\alpha$. In our experiments, we use

**Algorithm 1** RelPSI $H_{0,i} : D(P_{\hat{j}}, R) \geq D(P_i, R) \mid P_{\hat{j}}$ is selected.

1: **procedure** RELPSI$(\mathcal{M}, R, \alpha)$
2:     Estimate $\hat{\boldsymbol{\Sigma}}$ given in Theorem C.1 and Theorem C.2 for KSD and MMD respectively.
3:     $\boldsymbol{r} \leftarrow (0, \ldots, 0) \in \{0, 1\}^l$
4:     $\boldsymbol{z} \leftarrow [\sqrt{n}\hat{D}(P_1, R), \sqrt{n}\hat{D}(P_2, R), \ldots, \sqrt{n}\hat{D}(P_l, R)]^\top$
5:     $\hat{J} \leftarrow \arg\min_{j \in \mathcal{I}} \hat{D}(P_j, R)$.
6:     Compute $\boldsymbol{A}$ and $\boldsymbol{b}$ (as defined in Section 3.1).
7:     **for** $i \in \mathcal{I} : i \neq \hat{J}$ **do**
8:         $\boldsymbol{\eta} \leftarrow [0, \ldots, \overbrace{-1}^{\hat{j}}, \ldots, \overbrace{1}^{i}, \ldots, 0]^\top$
9:         $\hat{\sigma} \leftarrow \sqrt{\boldsymbol{\eta}^\top \hat{\boldsymbol{\Sigma}} \boldsymbol{\eta}}$
10:        Compute $\mathcal{V}^+$ and $\mathcal{V}^-$ (described in Lemma 3.1).
11:        $\hat{t}_\alpha \leftarrow \hat{\sigma}\Phi^{-1}\left((1-\alpha)\Phi\left(\frac{\mathcal{V}^+}{\hat{\sigma}}\right) + \alpha\Phi\left(\frac{\mathcal{V}^-}{\hat{\sigma}}\right)\right)$
12:        $\boldsymbol{r}_i \leftarrow \boldsymbol{\eta}^\top \boldsymbol{z} > \hat{t}_\alpha$
13:     **end for**
14: **return** $\boldsymbol{r}$
15: **end procedure**

---

**Algorithm 2** RelMulti $H_{0,i} : D(P_{\hat{j}}, R) \geq D(P_i, R) \mid P_{\hat{j}}$ is selected.

1: **procedure** RELMULTI$(\mathcal{M}, R, \alpha, \rho)$
2:     Estimate $\hat{\boldsymbol{\Sigma}}$ as given in Theorem C.1 and Theorem C.2 for KSD and MMD respectively.
3:     $\mathcal{D}_0, \mathcal{D}_1 \leftarrow$ SplitData$(\mathcal{M}, R, \rho)$
4:     $n_1 \leftarrow \rho n$
5:     (With $\mathcal{D}_0$) $\hat{J} \leftarrow \arg\min_{j \in \mathcal{I}} \hat{D}(P_j, R)$.
6:     Compute $\boldsymbol{A}$ and $\boldsymbol{b}$.
7:     **for** $i \in \mathcal{I} : i \neq \hat{J}$ **do** (with $\mathcal{D}_1$)
8:         Compute $\boldsymbol{z}_2 = [\sqrt{n_1}\hat{D}(P_1, R), \sqrt{n_1}\hat{D}(P_2, R), \ldots, \sqrt{n_1}\hat{D}(P_l, R)]^\top$
9:         $\boldsymbol{\eta}^\top = [0, \ldots, \overbrace{-1}^{\hat{j}} \ldots, \overbrace{1}^{i}, \ldots, 0]$
10:        $\hat{\sigma} \leftarrow \sqrt{\boldsymbol{\eta}^\top \hat{\boldsymbol{\Sigma}} \boldsymbol{\eta}}$
11:        $\boldsymbol{p}_i \leftarrow 1 - \Phi(\frac{\boldsymbol{\eta}^\top \boldsymbol{z}_2}{\hat{\sigma}})$
12:     **end for**
13: **return** FDRCorrection$(\boldsymbol{p}, \alpha)$
14: **end procedure**

---

the Benjamini–Yekutieli procedure [1]. SplitData$(\mathcal{M}, R, \rho)$ is a function that splits the samples generated by $R$ and $\mathcal{M}$ (if it is represented by samples). It returns two datasets $\mathcal{D}_0$ and $\mathcal{D}_1$ such that $|\mathcal{D}_0| = (1 - \rho)n$ and $|\mathcal{D}_1| = \rho n$.

## C   Asymptotic distributions

In this section, we prove the asymptotic distribution of $\widehat{\mathrm{KSD}}_u^2$ and also provide the asymptotic distribution of $\widehat{\mathrm{MMD}}_u^2$ for completeness.

**Theorem C.1** (Asymptotic Distribution of $\widehat{\mathrm{KSD}_u^2}$)**.** *Let $P_i, P_j$ be distributions with density functions $p_i, p_j$ respectively, and let $R$ be the data generating distribution. Assume that $P_i, P_j, R$ are distinct. We denote a sample by $Z_n = Z \sim R$.* $\sqrt{n}\left(\begin{pmatrix} \widehat{\mathrm{KSD}}_u^2(P_i, Z) \\ \widehat{\mathrm{KSD}}_u^2(P_j, Z) \end{pmatrix} - \begin{pmatrix} \mathrm{KSD}^2(P_i, R) \\ \mathrm{KSD}^2(P_j, R) \end{pmatrix}\right) \xrightarrow{d} \mathcal{N}(\boldsymbol{0}, \boldsymbol{\Sigma})$,

*where* $\mathbf{\Sigma} = \begin{pmatrix} \sigma^2_{P_i R} & \sigma_{P_i R P_j R} \\ \sigma_{P_i R P_j R} & \sigma^2_{P_j R} \end{pmatrix}$, $\sigma_{P_i R P_j R} = \mathrm{Cov}_{x \sim R}[\mathbb{E}_{x' \sim R}[u_{p_i}(x, x')], \mathbb{E}_{x' \sim R}[u_{p_j}(x, x')]]$ *and* $\sigma^2_{P_i R} = \mathrm{Var}_{x' \sim R}[\mathbb{E}_{x' \sim R}[u_{p_i}(x, x')]$.

*Proof.* Let $\mathcal{X} = \{x_i\}_{i=1}^n$ be $n$ i.i.d. random variables drawn from $R$ and we have a model with its corresponding gradient of its log density $s_{P_i}(x) = \nabla_x \log p_i(x)$. The complete U-statistic estimate of KSD between $P_i$ and $R$ is

$$\widehat{\mathrm{KSD}}^2_u(P_i, R) = \mathbb{E}_{x, x' \sim R}[u_{p_i}(x, x')] \approx \frac{1}{n_2} \sum_{i \neq j}^n u_{p_i}(x_i, x_j)$$

where $u_{p_i}(x, y) = s_{p_i}(x)^\top s_{p_i}(y) k(x, y) + s_{p_i}(y)^\top \nabla_x k(x, y) + s_{p_i}(x)^\top \nabla_y k(x, y) + \mathrm{tr}[\nabla_{x,y} k(x, y)]$ and $n_2 = n(n - 1)$.

Similarly, for another model $P_j$ and its gradient of its log density $s_{P_j}(x) = \nabla_x \log p_j(x)$. Its estimator is

$$\widehat{\mathrm{KSD}}^2_u(P_j, R) = \mathbb{E}_{x, x' \sim R}[u_{p_j} x, x')] \approx \frac{1}{n_2} \sum_{i \neq j}^n u_{p_j}(x_i, x_j)$$

where $u_{p_j}(x, y) = s_{p_j}(x)^\top s_{p_j}(y) k(x, y) + s_{p_j}(y)^\top \nabla_x k(x, y) + s_{p_j}(x)^\top \nabla_y k(x, y) + \mathrm{tr}[\nabla_{x,y} k(x, y)]$.

The covariance matrix of a U-statistic with a kernel of order 2 is

$$\mathbf{\Sigma} = \frac{4(n - 2)}{n(n - 1)} \zeta + \mathcal{O}_p(n^{-2})$$

where, for the variance term and covariance term, we have $\zeta_{ii} = \mathrm{Var}_{x \sim R}(\mathbb{E}_{y \sim R}[u_{p_i}(x, y)])$ and $\zeta_{ij} = \mathrm{Cov}_{x \sim R}(\mathbb{E}_{y \sim R}[u_{p_i}(x, y)], \mathbb{E}_{y \sim R}[u_{p_j}(x, y)])$ respectively. $\qquad \square$

The asymptotic distribution is provided below and is shown to be the case by Bounliphone et al. [4].

**Theorem C.2** (Asymptotic Distribution of $\widehat{\mathrm{MMD}}^2_u$ [4]). *Assume that $P_i$, $P_j$ and $R$ are distinct. We denote samples $X \sim P_i$, $Y \sim P_j$, $Z \sim R$.*

$$\sqrt{n}\left( \begin{pmatrix} \widehat{\mathrm{MMD}}^2_u(P_i, Z) \\ \widehat{\mathrm{MMD}}^2_u(P_j, Z) \end{pmatrix} - \begin{pmatrix} \mathrm{MMD}^2(P_i, R) \\ \mathrm{MMD}^2(P_j, R) \end{pmatrix} \right) \xrightarrow{d} \mathcal{N}(\mathbf{0}, \mathbf{\Sigma})$$

*where* $\mathbf{\Sigma} = \begin{pmatrix} \sigma^2_{P_i R} & \sigma_{P_i R P_j R} \\ \sigma_{P_i R P_j R} & \sigma^2_{P_j R} \end{pmatrix}$ *and* $\sigma_{P_i R P_j R} = \mathrm{Cov}[\mathbb{E}_{x' \sim P_i \times R}[h(X, x')], \mathbb{E}_{x' \sim P_j \times R}[g(X, x')]]$ *and* $\sigma^2_{P_j R} = \mathrm{Var}[\mathbb{E}_{x' \sim P_j \times R}[h(X, x')]$.

## D  Relative Kernelized Stein Discrepancy (RelKSD)

In this section, we describe the testing procedure for relative tests with KSD (a simple extension of RelMMD [4]). Currently, there is no test of relative fit with Kernelized Stein Discrepancy, and so we propose such a test using the complete estimator $\widehat{\mathrm{KSD}}^2_u$ which we call RelKSD. The test mirrors the proposal of Bounliphone et al. [4] and, given the asymptotic distribution of $\widehat{\mathrm{KSD}}^2_u$, it is a simple extension since its cross-covariance is known (see Theorem C.1).

Given two candidate models $P_1$ and $P_2$ with a reference distribution $R$ with its samples denoted as $Z \sim R$, we define our test statistic as $\sqrt{n}[\widehat{\mathrm{KSD}}^2_u(P_1, Z) - \widehat{\mathrm{KSD}}^2_u(P_2, Z)]$. For the test of relative similarity, we assume that the candidate models ($P_1$ and $P_2$) and unknown generating distribution $R$ are all distinct. Then, under the null hypothesis $H_0 : \mathrm{KSD}^2(P_1, R) - \mathrm{KSD}^2(P_2, R) \leq 0$, we can derive the asymptotic null distribution as follows. By the continuous mapping theorem and Theorem C.1, we have

$$\sqrt{n}[\widehat{\mathrm{KSD}}^2_u(P_1, Z) - \widehat{\mathrm{KSD}}^2_u(P_2, Z)] - \sqrt{n}[\mathrm{KSD}^2(P_1, R) - \mathrm{KSD}^2(P_2, R)] \xrightarrow{d} \mathcal{N}\left(0, \sigma^2\right)$$

where $\sigma^2 = \begin{pmatrix} 1 \\ -1 \end{pmatrix}^\top \Sigma \begin{pmatrix} 1 \\ -1 \end{pmatrix} = \sigma^2_{P_1 R} - 2\sigma_{P_1 R P_2 R} + \sigma^2_{P_2 R}$, and $\Sigma$ is defined in Theorem C.1 (which we assume is positive definite). We will also use the most conservative threshold by letting the rejection threshold $t_\alpha$ be the $(1 - \alpha)$-quantile of the asymptotic distribution of $\sqrt{n}[\widehat{\text{KSD}}_u^2(P_1, Z) - \widehat{\text{KSD}}_u^2(P_2, Z)]$ with mean zero (see Bounliphone et al.[4]). If our statistic is above the $t_\alpha$, we reject the null.

# E   Calibration of the test

In this section, we will show that the p-values obtained are well calibrated, when two distributions are equal, measured by either MMD or KSD. The distribution of p-values should be uniform. Figure 3 shows the empirical CDF of p-values and should lie on the line if it is calibrated. Additionally, we show the empirical distribution of p values for a three of different mean shift problems where observed distribution is $R = \mathcal{N}(0, 1)$ and our candidate models are $P_1 = \mathcal{N}(\mu_1, 1)$ and $P_2 = \mathcal{N}(\mu_2, 1)$

Figure 3: Mean shift experiment described in experiments with (a) $\mu_1 = 0.5$ and $\mu_2 = -0.5$, (b) $\mu_1 = 2.5$ and $\mu_2 = 2.5$, (c) $\mu_1 = 2.5$ and $\mu_2 = -2.5$.

# F   Performance analysis for two models

In this section, we analyse the performance of our two proposed methods: RelPSI and RelMulti for $l = 2$ candidate models. We begin by computing the probability that we select the best model correctly (and selecting incorrectly). Then provide a closed form formula for computing the rejection threshold, and from this we were able to characterize the probability of rejection and proof our theoretical result.

## F.1   Cumulative distribution function of a truncated normal

The cumulative distribution function (CDF) of a truncated normal is given by

$$\Psi(x \mid \mu, \sigma, \mathcal{V}^-, \mathcal{V}^+) = \frac{\Phi(\frac{x-\mu}{\sigma}) - \Phi(\frac{\mathcal{V}^- - \mu}{\sigma})}{\Phi(\frac{\mathcal{V}^+ - \mu}{\sigma}) - \Phi(\frac{\mathcal{V}^- - \mu}{\sigma})},$$

where $\Phi$ is the CDF of the standard normal distribution [5, Section 3.3].

### F.2 Characterizing the selection event

Under the null and alternative hypotheses, for both $\widehat{\text{MMD}}_u^2(P, R)$ and $\widehat{\text{KSD}}_u^2(P, R)$, the test statistic is asymptotically normal i.e., for a sufficiently large $n$, we have

$$\sqrt{n}\big[\hat{D}(P_2, R) - \hat{D}(P_1, R) - \mu\big] \sim \mathcal{N}(0, \sigma^2),$$

where $\mu := D(P_2, R) - D(P_1, R)$ is the population difference and $D(\cdot, \cdot)$ can be either $\text{MMD}^2$ or $\text{KSD}^2$. The probability of selecting the model $P_1$, i.e., $P_{\hat{j}} = P_1$, is equivalent to the probability of observing $\hat{D}(P_1, R) < \hat{D}(P_2, R)$. The following lemma derives this quantity.

**Lemma F.1.** *Given two models $P_1$ and $P_2$, and the test statistic $\sqrt{n}[\hat{D}(P_2, R) - \hat{D}(P_1, R)]$ such that $\sqrt{n}\big[\hat{D}(P_2, R) - \hat{D}(P_1, R) - \mu\big] \xrightarrow{d} \mathcal{N}(0, \sigma^2)$, where $\mu := D(P_2, R) - D(P_1, R)$, then the probability that we select $P_1$ as the reference is*

$$\mathbb{P}(P_{\hat{j}} = P_1) = \mathbb{P}(\hat{D}(P_1, R) < \hat{D}(P_2, R)) \approx \Phi\left(\frac{\sqrt{n}\mu}{\sigma}\right).$$

*It follows that $\mathbb{P}(P_{\hat{j}} = P_2) \approx \Phi(-\frac{\sqrt{n}\mu}{\sigma})$.*

*Proof.* For some sufficiently large $n$, we have

$$\begin{aligned}
\mathbb{P}(P_{\hat{j}} = P_1) &= \mathbb{P}(\sqrt{n}\hat{D}(P_1, R) < \sqrt{n}\hat{D}(P_2, R)) \\
&= \mathbb{P}(\sqrt{n}[\hat{D}(P_2, R) - \hat{D}(P_1, R)] > 0) \\
&\approx 1 - \Phi(-\frac{\sqrt{n}\mu}{\sigma}) = \Phi\left(\frac{\sqrt{n}\mu}{\sigma}\right),
\end{aligned}$$

and $\mathbb{P}(P_{\hat{j}} = P_2) = 1 - \mathbb{P}(P_{\hat{j}} = P_1) \approx \Phi(-\frac{\sqrt{n}\mu}{\sigma})$. $\qquad\square$

It can be seen that as $n$ gets larger the selection procedure is more likely to select the correct model.

### F.3 Truncation points of RelPSI

To study the performance of RelPSI, it is necessary to characterize the truncation points in the polyhedral lemma (Theorem 3.1). In the case of two candidate models, the truncation points are simple as shown in Lemma F.2.

**Lemma F.2.** *Consider two candidate models $P_1$ and $P_2$ and the selection algorithm described in Section 3.1, with the test statistic $\sqrt{n}[\hat{D}(P_2, R) - \hat{D}(P_1, R)]$. The upper truncation point $\mathcal{V}^+$ and lower truncation point $\mathcal{V}^-$ (see Theorem 3.1) when the selection procedure observes $\hat{D}(P_1, R) < \hat{D}(P_2, R)$, i.e., $P_{\hat{j}} = P_1$, are*

$$\mathcal{V}^- = 0, \; \mathcal{V}^+ = \infty.$$

*When the selection procedure observes $\hat{D}(P_2, R) < \hat{D}(P_1, R)$, i.e., $P_{\hat{j}} = P_2$, then the truncation points are*

$$\mathcal{V}^- = -\infty, \; \mathcal{V}^+ = 0.$$

*Proof.* If the selection procedure observes that $\hat{D}(P_1, R) < \hat{D}(P_2, R)$ then $\hat{J} = 1$ and our test statistic is $\sqrt{n}[\hat{D}(P_2, R) - \hat{D}(P_{\hat{j}}, R)] = \sqrt{n}[\hat{D}(P_2, R) - \hat{D}(P_1, R)] = \boldsymbol{\eta}^\top \boldsymbol{z}$ where $\boldsymbol{\eta} = (-1 \quad 1)^\top$ and $\boldsymbol{z} = \sqrt{n}\begin{pmatrix} \hat{D}(P_1, R) \\ \hat{D}(P_2, R) \end{pmatrix}$. Then the affine selection event can be written as $\boldsymbol{Az} \leq \boldsymbol{b}$ where $\boldsymbol{A} = (1 \quad -1)$ and $\boldsymbol{b} = 0$. It follows from the definition of $\mathcal{V}^+$ and $\mathcal{V}^-$ (see Theorem 3.1) that we have $\mathcal{V}^- = 0$ and $\mathcal{V}^+ = \infty$.

A similar result holds for the case where the selection event observes $\hat{D}(P_2, R) < \hat{D}(P_1, R)$ (i.e., $\hat{J} = 2$). The test statistic is $\sqrt{n}[\hat{D}(P_1, R) - \hat{D}(P_2, R)]$. The selection event can be described with $\boldsymbol{A} = (-1 \quad 1)$ and $\boldsymbol{b} = 0$. Following from their definitions, we have $\mathcal{V}^- = -\infty$ and $\mathcal{V}^+ = 0$. $\qquad\square$

## F.4 Test threshold

Given a significance level $\alpha \in (0, 1)$, the test threshold is defined to the $(1 - \alpha)$-quantile of the truncated normal for RelPSI, and normal for RelMulti. The test threshold of the RelPSI is

$$t^{\text{RelPSI}}(\alpha) = \Psi^{-1}(1 - \alpha \,|\, \mu = 0, \sigma, \mathcal{V}^-, \mathcal{V}^+)$$

$$= \mu + \sigma \Phi^{-1}\left( (1 - \alpha)\Phi\left(\frac{\mathcal{V}^+ - \mu}{\sigma}\right) + \alpha \Phi\left(\frac{\mathcal{V}^- - \mu}{\sigma}\right) \right)$$

$$= \sigma \Phi^{-1}\left( (1 - \alpha)\Phi\left(\frac{\mathcal{V}^+}{\sigma}\right) + \alpha \Phi\left(\frac{\mathcal{V}^-}{\sigma}\right) \right),$$

where $\Psi^{-1}(\cdot \,|\, \mu, \sigma, \mathcal{V}^-, \mathcal{V}^+)$ is the inverse of the CDF of the truncated normal with mean $\mu$, standard deviation $\sigma$, and lower and upper truncation points denoted $\mathcal{V}^-, \mathcal{V}^+$, and $\Phi^{-1}$ is the inverse of the CDF of the standard normal distribution. Note that under the null hypothesis, $\mu \le 0$ (recall $\mu := D(P_2, R) - D(P_1, R)$), we set $\mu = 0$ which results in a more conservative test for rejecting the null hypothesis. Furthermore, we generally do not know $\sigma$; instead we use its plug-in estimator $\hat{\sigma}$.

Given two candidate models $P_1$ and $P_2$, the truncation points $(\mathcal{V}^-, \mathcal{V}^+)$ are either (see Lemma F.2):

- Case 1: $\mathcal{V}^- = 0$, $\mathcal{V}^+ = \infty$, or
- Case 2: $\mathcal{V}^- = -\infty$, $\mathcal{V}^+ = 0$.

The two cases result in different level-$\alpha$ rejection thresholds since the value of the rejection threshold is dependent on the truncation points.

For Case 1, the threshold is

$$t_1^{\text{RelPSI}}(\alpha) = \hat{\sigma}\Phi^{-1}\left(1 - \frac{\alpha}{2}\right).$$

For Case 2, the threshold is

$$t_2^{\text{RelPSI}}(\alpha) = \hat{\sigma}\Phi^{-1}\left(\frac{1}{2} - \frac{\alpha}{2}\right).$$

Note that since $\Phi^{-1}(\cdot)$ is monotonically increasing, we have $t_2^{\text{RelPSI}}(\alpha) < 0 < t_1^{\text{RelPSI}}(\alpha)$.

For RelMulti, the threshold is given by the $(1 - \alpha)$-quantile of the asymptotic null distribution which is a normal distribution (with the mean $\mu$ adjusted to 0):

$$t^{\text{RelMulti}}(\alpha) = \hat{\sigma}\Phi^{-1}(1 - \alpha).$$

## F.5 Rejection probability

Consider the test statistic $\sqrt{n}\hat{\mu} := \sqrt{n}\big[\hat{D}(P_2, R) - \hat{D}(P_1, R)\big]$.

**RelPSI**  Depending on whether $\hat{J} = 1$ or $\hat{J} = 2$, the rejection probability for RelPSI is given by

$$\mathbb{P}(\sqrt{n}\hat{\mu} > t_1^{\text{RelPSI}}(\alpha) \,|\, P_{\hat{j}} = P_1) \text{ or}$$

$$\mathbb{P}(\sqrt{n}\hat{\mu} > t_2^{\text{RelPSI}}(\alpha) \,|\, P_{\hat{j}} = P_2).$$

Assume $n$ is sufficiently large. The rejection probability of RelPSI when $P_{\hat{j}} = P_1$ is

$$\mathbb{P}(\sqrt{n}\hat{\mu} > t_1^{\text{RelPSI}}(\alpha) \,|\, P_{\hat{j}} = P_1) \approx 1 - \frac{\Phi\left(\frac{\hat{\sigma}\Phi^{-1}(1 - \frac{\alpha}{2}) - \sqrt{n}\mu}{\sigma}\right) - \Phi\left(\frac{-\sqrt{n}\mu}{\sigma}\right)}{1 - \Phi\left(\frac{-\sqrt{n}\mu}{\sigma}\right)}$$

$$\overset{(*)}{\approx} 1 - \frac{\Phi\left(\Phi^{-1}(1 - \frac{\alpha}{2}) - \frac{\sqrt{n}\mu}{\sigma}\right) - \Phi\left(\frac{-\sqrt{n}\mu}{\sigma}\right)}{1 - \Phi\left(\frac{-\sqrt{n}\mu}{\sigma}\right)}. \qquad (1)$$

When $P_{\hat{j}} = P_2$, it is

$$\mathbb{P}(\sqrt{n}\hat{\mu} > t_2^{\text{RelPSI}}(\alpha) \,|\, P_{\hat{j}} = P_2) \approx 1 - \frac{\Phi\left(\frac{\hat{\sigma}\Phi^{-1}(\frac{1}{2} - \frac{\alpha}{2}) - \sqrt{n}\mu}{\sigma}\right)}{\Phi\left(\frac{-\sqrt{n}\mu}{\sigma}\right)}$$

$$\overset{(*)}{\approx} 1 - \frac{\Phi\left(\Phi^{-1}(\frac{1}{2} - \frac{\alpha}{2}) - \frac{\sqrt{n}\mu}{\sigma}\right)}{\Phi\left(\frac{-\sqrt{n}\mu}{\sigma}\right)}. \qquad (2)$$

**RelMulti**   For RelMulti, it is $\mathbb{P}(\sqrt{n}\hat{\mu} > t^{\mathrm{RelMulti}}(\alpha))$. The rejection probability of RelMulti is

$$\mathbb{P}(\sqrt{n}\hat{\mu} > t^{\mathrm{RelMulti}}(\alpha)) \approx 1 - \Phi(\frac{\hat{\sigma}\Phi^{-1}(1-\alpha) - \sqrt{n}\mu}{\sigma})$$

$$\overset{(*)}{\approx} 1 - \Phi\left(\Phi^{-1}(1-\alpha) - \frac{\sqrt{n}\mu}{\sigma}\right), \tag{3}$$

where we use the fact that $\sqrt{n}(\hat{\mu} - \mu) \overset{d}{\to} \mathcal{N}(0, \sigma^2)$. We note that at $(*)$ we use the fact that as $n \to \infty$, $\hat{\sigma}$ converges to $\sigma$ in probability.

### F.6   True positive rates of RelPSI and RelMulti

For the remainder of the section, we assume without loss of generality that $D(P_1, R) < D(P_2, R)$, i.e., $P_1$ is the better model, so we have $\mu = D(P_2, R) - D(P_1, R) > 0$.

**RelPSI**   The TPR (for RelPSI) is given by

$$\mathrm{TPR}_{\mathrm{RelPSI}} = \mathbb{E}\left[\frac{\text{Number of True Positives assigned Positive}}{\underbrace{\text{Number of True Positives}}_{=1}}\right]$$

$$\overset{(a)}{=} \mathbb{P}(\text{decide that } P_2 \text{ is worse})$$
$$= \mathbb{P}(\text{decide that } P_2 \text{ is worse} \mid P_1 \text{ is selected})\mathbb{P}(P_1 \text{ is selected})$$
$$\quad + \mathbb{P}(\text{decide that } P_2 \text{ is worse} \mid P_2 \text{ is selected})\mathbb{P}(P_2 \text{ is selected})$$

$$\overset{(b)}{=} \mathbb{P}(\text{decide that } P_2 \text{ is worse} \mid P_1 \text{ is selected})\mathbb{P}(P_1 \text{ is selected})$$
$$= \mathbb{P}(\text{Reject } H_0 : D(P_1, R) \geq D(P_2, R) \mid P_1 \text{ is selected})\mathbb{P}(P_1 \text{ is selected})$$
$$= \mathbb{P}(\sqrt{n}\hat{\mu} > t_1^{\mathrm{RelPSI}}(\alpha) \mid P_{\hat{j}} = P_1)P(P_{\hat{j}} = P_1),$$

where we note that at $(a)$, deciding that $P_2$ is worse than $P_1$ is the same as assigning positive to $P_2$. The equality at $(b)$ holds due to the design of our procedure that only tests the selected reference against other candidate models to decide whether they are worse than the reference model. By design, we will not test the selected reference model against itself. So, $\mathbb{P}(\text{decide that } P_2 \text{ is worse} \mid P_2 \text{ is selected}) = 0$. Using Equation 1 and Lemma F.1, we have

$$\mathrm{TPR}_{\mathrm{RelPSI}} \approx \left[1 - \frac{\Phi(\Phi^{-1}(1-\frac{\alpha}{2}) - \frac{\sqrt{n}\mu}{\sigma}) - \Phi(\frac{-\sqrt{n}\mu}{\sigma})}{1 - \Phi(\frac{-\sqrt{n}\mu}{\sigma})}\right]\left[1 - \Phi(-\frac{\sqrt{n}\mu}{\sigma})\right]$$

$$= 1 - \Phi(-\frac{\sqrt{n}\mu}{\sigma}) - \Phi(\Phi^{-1}(1-\frac{\alpha}{2}) - \frac{\sqrt{n}\mu}{\sigma}) + \Phi(\frac{-\sqrt{n}\mu}{\sigma})$$

$$= 1 - \Phi(\Phi^{-1}(1-\frac{\alpha}{2}) - \frac{\sqrt{n}\mu}{\sigma})$$

$$= \Phi\left(\frac{\sqrt{n}\mu}{\sigma} - \Phi^{-1}(1-\frac{\alpha}{2})\right). \tag{4}$$

**RelMulti**   For RelMulti, we perform data splitting to create independent sets of our data for testing and selection. Suppose we have $n$ samples and a proportion of samples to be used for testing $\rho \in (0, 1)$, we have $m_1 = \rho n$ samples used for testing and $m_0 = n(1-\rho)$ samples for selection. Then TPR for RelMulti can be derived as

$$\mathrm{TPR}_{\mathrm{RelMulti}} = \mathbb{P}(\text{decide that } P_2 \text{ is worse} \mid P_1 \text{ is selected})\mathbb{P}(P_1 \text{ is selected})$$
$$= \mathbb{P}(\text{decide that } P_2 \text{ is worse})\mathbb{P}(P_1 \text{ is selected})$$
$$= \mathbb{P}(\sqrt{m_1}\hat{\mu} > t^{\mathrm{RelMulti}}(\alpha))\mathbb{P}(\sqrt{m_0}\hat{\mu} > 0).$$

Using Equation 3 (with $n\rho$ samples) and Lemma F.1 (with $n(1-\rho)$ samples), we have

$$\mathrm{TPR}_{\mathrm{RelMulti}} \approx \left[1 - \Phi(\Phi^{-1}(1-\alpha) - \frac{\sqrt{n\rho}\mu}{\sigma})\right]\Phi(\frac{\sqrt{n(1-\rho)}\mu}{\sigma}). \tag{5}$$

We note that both $\text{TPR}_{\text{RelPSI}} \to 1$ and $\text{TPR}_{\text{RelMulti}} \to 1$, as $n \to \infty$. We are ready to prove Theorem 4.1. We first recall the theorem from the main text:

**Theorem 4.1** (TPR of RelPSI and RelMulti). *Let $P_1, P_2$ be two candidate models, and $R$ be a data generating distribution. Assume that $P_1, P_2$ and $R$ are distinct. Given $\alpha \in [0, \frac{1}{2}]$ and split proportion $\rho \in (0,1)$ for RelMulti so that $(1-\rho)n$ samples are used for selecting $P_{\hat{j}}$ and $\rho n$ samples for testing, for all $n \gg N = \left( \frac{\sigma \Phi^{-1}(1-\frac{\alpha}{2})}{\mu(1-\sqrt{\rho})} \right)^2$, we have $\text{TPR}_{\text{RelPSI}} \gtrsim \text{TPR}_{\text{RelMulti}}$.*

*Proof.* Assume without loss of generality that $D(P_1, R) < D(P_2, R)$, i.e., $P_1$ is the best model, and $\mu := D(P_2, R) - D(P_1, R) > 0$ is the population difference of two discrepancy measures (which can be either MMD or KSD) and $\sigma$ is the standard deviation of our test statistic. Since $n > N$, we have

$$\frac{\sqrt{n}\mu}{\sigma}(1 - \sqrt{\rho}) \geq \Phi^{-1}(1 - \frac{\alpha}{2})$$

$$\overset{(a)}{\Longrightarrow} \frac{\sqrt{n}\mu}{\sigma} - \frac{\sqrt{n\rho}\mu}{\sigma} \geq \Phi^{-1}(1 - \frac{\alpha}{2}) - \overbrace{\Phi^{-1}(1 - \alpha)}^{\geq 0}$$

$$\equiv \frac{\sqrt{n}\mu}{\sigma} - \Phi^{-1}(1 - \frac{\alpha}{2}) \geq \frac{\sqrt{n\rho}\mu}{\sigma} - \Phi^{-1}(1 - \alpha)$$

$$\overset{(b)}{\Longrightarrow} \Phi\left( \frac{\sqrt{n}\mu}{\sigma} - \Phi^{-1}(1 - \frac{\alpha}{2}) \right) \geq \Phi\left( \frac{\sqrt{n\rho}\mu}{\sigma} - \Phi^{-1}(1 - \alpha) \right),$$

where at $(a)$, we have $\Phi^{-1}(1 - \alpha) \geq 0$ because $\alpha \in [0, 1/2]$. At $(b)$, we use the fact that $a \mapsto \Phi(a)$ is increasing. We note that the left hand side is the same as Equation (4) and it follows that

$$\text{TPR}_{\text{RelPSI}} \gtrsim \Phi\left( \frac{\sqrt{n\rho}\mu}{\sigma} - \Phi^{-1}(1 - \alpha) \right) \underbrace{\Phi(\frac{\sqrt{n(1-\rho)}\mu}{\sigma})}_{\in (0,1)}$$

$$= \left[ 1 - \Phi(\Phi^{-1}(1 - \alpha) - \frac{\sqrt{n\rho}\mu}{\sigma}) \right] \left[ \Phi(\frac{\sqrt{n(1-\rho)}\mu}{\sigma}) \right]$$

$$\overset{(c)}{\gtrsim} \text{TPR}_{\text{RelMulti}},$$

where at $(c)$ we use Equation (5). $\qquad\square$

## G   Test consistency

In this section, we describe and prove the consistency result of our proposal RelPSI and RelMulti for both MMD and KSD.

**Theorem 3.2** (Consistency of RelPSI-MMD). *Given two models $P_1$, $P_2$ and a data distribution $R$ (which are all distinct). Let $\hat{\Sigma}$ be a consistent estimate of the covariance matrix defined in Theorem C.2. and $\boldsymbol{\eta}$ be defined such that $\boldsymbol{\eta}^\top \boldsymbol{z} = \sqrt{n}[\widehat{\text{MMD}}_u^2(P_2, R) - \widehat{\text{MMD}}_u^2(P_1, R)]$. Suppose that the threshold $\hat{t}_\alpha$ is the $(1-\alpha)$-quantile of $\mathcal{TN}(0, \boldsymbol{\eta}^\top \hat{\Sigma} \boldsymbol{\eta}, \mathcal{V}^-, \mathcal{V}^+)$ where $\mathcal{V}^+$ and $\mathcal{V}^-$ are defined in Theorem 3.1. Under $H_0 : \boldsymbol{\eta}^\top \boldsymbol{\mu} \leq 0 \,|\, P_{\hat{j}}$ is selected, the asymptotic type-I error is bounded above by $\alpha$. Under $H_1 : \boldsymbol{\eta}^\top \boldsymbol{\mu} > 0 \,|\, P_{\hat{j}}$ is selected, we have $\mathbb{P}(\boldsymbol{\eta}^\top \boldsymbol{z} > \hat{t}_\alpha) \to 1$ as $n \to \infty$.*

*Proof.* Let $\hat{t}_\alpha$ and $t_\alpha$ be $(1-\alpha)$ quantiles of distributions $\mathcal{TN}(0, \boldsymbol{\eta}^\top \hat{\Sigma} \boldsymbol{\eta}, \mathcal{V}^-, \mathcal{V}^+)$ and $\mathcal{TN}(0, \boldsymbol{\eta}^\top \Sigma \boldsymbol{\eta}, \mathcal{V}^-, \mathcal{V}^+)$ respectively. Given that $\hat{\Sigma} \overset{p}{\to} \Sigma$, $\mathcal{TN}(0, \boldsymbol{\eta}^\top \hat{\Sigma} \boldsymbol{\eta}, \mathcal{V}^-, \mathcal{V}^+)$ converges to $\mathcal{TN}(0, \boldsymbol{\eta}^\top \Sigma \boldsymbol{\eta}, \mathcal{V}^-, \mathcal{V}^+)$ in probability, hence, $\hat{t}_\alpha$ converges to $t_\alpha$. Note that $\hat{t}_\alpha$ is random and is determined by which model is selected to be $P_{\hat{j}}$ (which changes the truncation points $\mathcal{V}^-$ and $\mathcal{V}^+$).

Under $H_0 : \boldsymbol{\eta}^\top \boldsymbol{\mu} \leq 0 \,|\, P_{\hat{j}}$ is selected, for some sufficiently large $n$ the rejection rate is

$$\lim_{n \to \infty} \mathbb{P}_{H_0}(\boldsymbol{\eta}^\top \boldsymbol{z} > \hat{t}_\alpha) = \lim_{n \to \infty} \mathbb{P}_{H_0}(\boldsymbol{\eta}^\top \boldsymbol{z} > t_1^{\text{RelPSI}}(\alpha) \,|\, P_{\hat{j}} = P_1) \mathbb{P}(P_{\hat{j}} = P_1)$$

$$+ \lim_{n \to \infty} \mathbb{P}_{H_0}(\boldsymbol{\eta}^\top \boldsymbol{z} > t_2^{\text{RelPSI}}(\alpha) | P_{\hat{j}} = P_2) \mathbb{P}(P_{\hat{j}} = P_2).$$

Using Lemma F.1 with Equation 1 and Equation 2, we have

$$\lim_{n\to\infty} \mathbb{P}_{H_0}(\boldsymbol{\eta}^\top \boldsymbol{z} > \hat{t}_\alpha) = 1 - \lim_{n\to\infty} \Phi(\Phi^{-1}(1 - \frac{\alpha}{2}) - \frac{\sqrt{n}\boldsymbol{\eta}^\top \boldsymbol{\mu}}{\sigma})$$

$$+ \lim_{n\to\infty} \Phi(-\frac{\sqrt{n}\boldsymbol{\eta}^\top \boldsymbol{\mu}}{\sigma}) - \lim_{n\to\infty} \Phi(\Phi^{-1}(\frac{1}{2} - \frac{\alpha}{2}) - \frac{\sqrt{n}\boldsymbol{\eta}^\top \boldsymbol{\mu}}{\sigma}).$$

$$\leq 1 - \Phi(\Phi^{-1}(1 - \frac{\alpha}{2}))$$

$$+ \frac{1}{2} - \Phi(\Phi^{-1}(\frac{1}{2} - \frac{\alpha}{2}))$$

$$\leq \alpha.$$

Under $H_1 : \boldsymbol{\eta}^\top \boldsymbol{\mu} > 0 \,|\, P_{\hat{j}}$ is selected, similarly to $H_0$ we have

$$\lim_{n\to\infty} \mathbb{P}_{H_1}(\boldsymbol{\eta}^\top \boldsymbol{z} > \hat{t}_\alpha) = \lim_{n\to\infty} \mathbb{P}_{H_1}(\boldsymbol{\eta}^\top \boldsymbol{z} > \hat{t}_\alpha | P_{\hat{j}} = P_1)\mathbb{P}(P_{\hat{j}} = P_1)$$

$$+ \lim_{n\to\infty} \mathbb{P}_{H_1}(\boldsymbol{\eta}^\top \boldsymbol{z} > \hat{t}_\alpha | P_{\hat{j}} = P_2)\mathbb{P}(P_{\hat{j}} = P_2)$$

$$= 1 - \lim_{n\to\infty} \Phi(\Phi^{-1}(1 - \frac{\alpha}{2}) - \frac{\sqrt{n}\boldsymbol{\eta}^\top \boldsymbol{\mu}}{\sigma})$$

$$+ \lim_{n\to\infty} \Phi(-\frac{\sqrt{n}\boldsymbol{\eta}^\top \boldsymbol{\mu}}{\sigma}) - \lim_{n\to\infty} \Phi(\Phi^{-1}(\frac{1}{2} - \frac{\alpha}{2}) - \frac{\sqrt{n}\boldsymbol{\eta}^\top \boldsymbol{\mu}}{\sigma}),$$

where $\boldsymbol{\eta}^\top \boldsymbol{\mu}$ is the population difference of the two discrepancy measures, and $\sigma$ the standard deviation.

Since the alternative hypothesis is true, i.e., $\boldsymbol{\eta}^\top \boldsymbol{\mu} > 0$, we have $\lim_{n\to\infty} \mathbb{P}_{H_1}(\boldsymbol{\eta}^\top \boldsymbol{z} > \hat{t}_\alpha) = 1$. □

**Theorem G.1** (Consistency of RelPSI-KSD). *Given two models $P_1$, $P_2$ and reference distribution $R$ (which are all distinct). Let $\hat{\boldsymbol{\Sigma}}$ be the covariance matrix defined in Theorem C.1 and $\boldsymbol{\eta}$ be defined such that $\boldsymbol{\eta}^\top \boldsymbol{z} = \sqrt{n}[\widehat{\mathrm{KSD}}_u^2(P_1, R) - \widehat{\mathrm{KSD}}_u^2(P_2, R)]$. Suppose that the threshold $\hat{t}_\alpha$ is the $(1 - \alpha)$-quantile of the distribution of $\mathcal{TN}(\mathbf{0}, \boldsymbol{\eta}^\top \hat{\boldsymbol{\Sigma}} \boldsymbol{\eta}, \mathcal{V}^-, \mathcal{V}^+)$ where $\mathcal{V}^+$ and $\mathcal{V}^-$ is defined in Theorem 3.1. Under $H_0 : \boldsymbol{\eta}^\top \boldsymbol{\mu} \leq 0 \,|\, P_{\hat{j}}$ is selected, the asymptotic type-I error is bounded above by $\alpha$. Under $H_1 : \boldsymbol{\eta}^\top \boldsymbol{\mu} > 0 \,|\, P_{\hat{j}}$ is selected, we have $\mathbb{P}(\boldsymbol{\eta}^\top \boldsymbol{z} > \hat{t}_\alpha) \to 1$ as $n \to \infty$.*

## H  Additional experiments

In this section, we show results of two experiments. The first investigates the behaviour of RelPSI and RelMulti for multiple candidate models; and the second focuses on empirically verifying the implication of Theorem 4.1.

### H.1  Multiple candidate models experiment

In the following experiments, we demonstrate our proposal for synthetic problems when there are more than two candidate models and report the empirical true positive rate $\widehat{\mathrm{TPR}}$, empirical false discovery rate $\widehat{\mathrm{FDR}}$, and empirical false positive rate $\widehat{\mathrm{FPR}}$. We consider the following problems:

1. *Mean shift ($l = 10$):* There are many candidate models that are equally good. We set nine models to be just as good, compared to the reference $R = \mathcal{N}(\mathbf{0}, \boldsymbol{I})$, with one model that is worse than all of them. To be specific, the set of equally good candidates are defined as $\mathcal{I}_- = \{\mathcal{N}(\boldsymbol{\mu}_i, \boldsymbol{I}) : \boldsymbol{\mu}_i \in \{[0.5, 0, \ldots, 0], [-0.5, 0, \ldots, 0], [0, 0.5, \ldots, 0], [0, -0.5, \ldots, 0], \ldots\}\}$ and for the worst model, we have $Q = \mathcal{N}([1, 0, \ldots, 0], \boldsymbol{I})$. Our candidate model list is defined as $\mathcal{M} = \mathcal{I}_- \cup \{Q\}$. Each model is defined on $\mathbb{R}^{10}$.

2. *Restricted Boltzmann Machine ($l = 7$):* This experiment is similar to Experiment 1. Each candidate model is a Gaussian Restricted Boltzmann Machine with different perturbations of the unknown RBM parameters (which generates our unknown distribution $R$). We show how the behaviour of our proposed test vary with the degree of perturbation $\epsilon$ of a single model while the rest of the candidate models remain the same. The perturbation changes the model

from the best to worse than the best. Specifically, we have $\epsilon \in \{0.18, 0.19, 0.20, 0.22\}$ and the rest of the six models have fixed perturbation of $\{0.2, 0.3, 0.35, 0.4, 0.45, 0.5\}$. This problem demonstrates the sensitivity of each test.

The results from the mean shift experiment are shown in Figure 4 and results from RBM experiment are shown in Figure 5. Both experiments show that $\widehat{FPR}$ and $\widehat{FDR}$ is controlled for RelPSI and RelMulti respectively. As before, KSD-based tests exhibits the highest $\widehat{TPR}$ in the RBM experiment. In the mean shift example, RelPSI has lower $\widehat{TPR}$ compared with RelMulti and is an example where condition on the selection event results in a lower $\widehat{TPR}$ (lower than data splitting). In both experiments, for RelMulti $50\%$ of the data is used for selection and $50\%$ for testing.

Figure 4: Mean Shift Experiment: Rejection rates (estimated from 300 trials) for the six tests with $\alpha = 0.05$ is shown.

Figure 5: RBM Experiment. Rejection rates (estimated from 300 trials) for the six tests with $\alpha = 0.05$ is shown.

## H.2 TPR experiment

For this experiment, our goal is to empirically evaluate and validate Theorem 4.1 where $l = 2$. For some sufficiently large $n$, it states that the TPR of RelPSI will be an upper bound for the TPR of RelMulti (for both MMD and KSD). We consider the following two synthetic problems:

1. *Mixture of Gaussian*: The candidate models and unknown distribution are 1-d mixture of Gaussians where $M(\rho) = \rho \mathcal{N}(1,1) + (1-\rho)\mathcal{N}(-1,1)$ with mixing portion $\rho \in (0,1)$. We set the reference to be $R = M(0.5)$, and two candidate models to $P_1 = M(0.7)$ and

Figure 6: $l = 2$: The empirical true positive rate $\widehat{\text{TPR}}$ of $\widehat{\text{MMD}}_u^2$ (a) for the mixture problem of (b). We show empirical $\widehat{\text{TPR}}$ of $\widehat{\text{KSD}}_u^2$ (c) for rotation problem of (d). S:$a$ T:$b$ indicates that $a\%$ of the original dataset is used for selection and $b\%$ of the dataset used for testing.

$P_2 = M(0.75)$ (see Figure 6b). In this case, $P_1$ is closer to the reference distribution but only by a small amount. In this problem, we apply MMD and report the behaviour of the test as $n$ increases.

2. *Rotating Gaussian:* The two candidate models and our reference distributions are 2-d Gaussian distributions that differ by rotation (see Figure 6d). We fix the sample size to $n = 500$. Instead, we rotate the Gaussian distribution $P_1$ away from $P_2$ such that $P_1$ continues to get closer to the reference $R$ with each rotation. They are initially the same distribution but $P_1$ becomes a closer relative fit (with each rotation). In this problem, we apply KSD and report the empirical TPR as the Gaussian rotates and becomes an easier problem.

For each problem we consider three possible splits of the data: $25\%$, $50\%$, $25\%$ of the original samples for selection (and the rest for testing). Both problems use a Gaussian kernel with bandwidth set to $1$. The overall results are shown in Figure 6.

In Figure 6a, we plot the $\widehat{\text{TPR}}$ for RelPSI-MMD and RelMulti-MMD for the Mixture of Gaussian problem. RelPSI performs the best with the highest empirical $\widehat{\text{TPR}}$ confirming with Theorem 4.1. The next highest is RelMulti that performs a S:$25\%$ T:$75\%$ selection test split. The worst performer is the RelMulti with S:$75\%$ T:$25\%$ selection test split which can be explained by noting that most of the data has been used in selection, there is an insufficient amount of remaining data points to reject the hypothesis. The same behaviour can be observed in Figure 6c for $\widehat{\text{KSD}}_u^2$. Overall, this experiment corroborates with our theoretical results that TPR of RelPSI will be higher in population.