[Reviews · NeurIPS 2019]

Reviewer 1



Overall I think this is a nice and very clearly written paper. The authors made effort to make the article both precise and readable. The extension of model comparison using KSD rather than MMD is, I think, significant as KSD can be sometimes a more appropriate choice of divergence than MMD. On the other hand I thought the authors didn't really justify why it was important to consider the more general case in which there were more than 2 models to compare: is this an important problem? While in the case l=2, the first experiment shows RelPSI does not perform better than Rel and loses power. A couple of minor comments : (Line) 90 - the kernel does not define an inner product: the RKHS has an inner product since it is a Hilbert space 93 - "we interchangeably write k(x,.) and \phi(x)": why is this justified? You havent specified which feature map \phi is 97 - MMD^2 is not a pseudometric 99 - "MMD defines a metric, i.e., MMD^2(P,Q) = 0 iff P=Q": that s not the definition of a metric 107 - original -> originally 109 - we need p>0 for the score to be defined (and the kernel needs to be differentiable in the next line) 111 - What norm is used on H^d? 115 - u_p is a matrix rather than a scalar since the gradient is a row vector by mathematical convention unless otherwise specified 116 - \hat{KSD}_{u}(P,R) should be just \hat{KSD}_{u}, and the x,x' in the sum are missing their indices 150 - should be argmin instead of min 179-180 - I didnt really understand this sentence

Reviewer 2



*** UPDATE *** Thank you for your response to the points raised. The modifications to the presentation that have been discussed should lead to an improvement in the manuscript, so I have increased my score by one level. This paper derives the distribution of test statistics based on maximum mean discrepancy (MMD) and kernel Stein discrepancy (KSD). The focus is on testing for the "best" model among a range of candidates, as opposed to goodness-of-fit testing. The use of these test statistics is empirically illustrated. My impression of this manuscript is that the idea and the analysis is probably fine - testing is not something in which I am an expert - but that the idea is somewhat incremental and the writing style comes across as sloppy and imprecise, to the overall detriment of the manuscript. Thus the comments below are an extensive list of presentational issues which need to be fixed: l1. The first sentence of the abstract, frustratingly, isn't clear. I think the way that the positive and negative labels are being assigned is the opposite to what seems natural, and that causes confusion. To me "positive" should naturally be associated with "joint best model". l58. "for data" -> "from data" l95. The authors write "(assumed to exist under mild conditions)". If interpreted literally, this would be ridiculous, so rephrasing is needed. Moreover, the authors should actually spell out the "mild conditions" and provide a reference. l131. The authors write "J = argmin" which is extremely confusing given that one of the central premises of the manuscript is that the "best" model need not be unique. The "\in" symbol should be used instead. Similarly on l150. l158. The symbol \mu is undefined. I would guess it is meant to be z. l231. The statement of Theorem 4.1 is unclear to me - I did not understand it. For example, TPR is a function of the random sample, so how can it be true that TPR_PSI >= TPR_Split holds deterministically for all N? Could there not be some realisation of the data such that TPR_PSI < TPR_Split? l234. Moreover, the statement of Theorem 4.1 is unacceptably imprecise. For example, \mu is defined in words as "the population difference of two discrepancy measures". Which discrepancy measures? The authors should be precise and use mathematical formula instead. l240. There are grammatical issues with Lemma 4.2 which preclude a precise mathematical interpretation of what is being said. These include, but are not limited to, l241 where "best. We" should be "best, we". (Also, see lA458.) l244. The authors reference equation 4.2, but there is no equation 4.2... l271. The authors write that "In this case, H0 is true". However, in this case P_1 and P_2 seem equally close to R, so why is H0 true? It would have thought it was false, since H0 would prefer P_1 to P_2. l287. The line "linear time estimators perform the worse than the complete estimators counterpart" has a grammatical issue, but also the issue that linear time estimators do not seem to be mentioned beforehand in the main text. l288. The results in Fig.1 do not seem to account for the sampling variability of the dataset, and seem to be based on a single realisation of the dataset. Some assessment of the sampling randomness on these results is needed. l444. "stein" -> "Stein" lA474. The authors call \nabla_x \log p_i(x) the "log density", but it is the gradient of the log density. lA477. One of the \nabla_x should be a \nabla_y. See also lA481. lA478. Is writing "m_2 = m(m-1) / 1" really necessary? (i.e. remove the "/ 1").

Reviewer 3



I thank the authors for their response, and would like to maintain my overall evaluation. ===== The paper is generally clear, with minor grammatical blemishes. The paper generalizes the relative goodness-of-fit tests for l = 2 models in [4, 17] to comparing l > 2 models by controlling FDR or FPR under the post-selection inference framework (along the lines of [20, 31]) and is in this sense somewhat incremental. On the other hand, the paper makes some reasonable (albeit not groundbreaking) theoretical contributions, and the experiments are fairly extensive. Several comments/questions: - Both of the proposed approaches rely on initially selecting a reference model. But what happens if among the list of candidate models, the chosen reference model (i.e., the model that minimizes the chosen discrepancy measure) is not in fact the model of best fit? This scenario could be quite plausible when the chosen discrepancy measure (e.g., inappropriate kernel bandwidth) or when the sample size is small. In this case, even if the latter testing procedures are consistent, one may never arrive at the correct conclusion. Consider, for instance, the toy example where three candidate models are normal distributions with the different means mu and standard deviation .1: A: mu = .9 B: mu = 1 C: mu = 1.2 and simulate n samples from model B. Then, it may be possible (due to noise in the data when n is small) that the reference model was mistakenly selected as model A rather than model B. In this case, it may be that the subsequent testing procedures would always reject model C since it is significantly different A, but model C might not have been rejected if model B had been selected as the reference model. In short, I feel that this approach of selecting an initial reference model and having all the subsequent testing procedure rely on the correctness of that initial choice seems a bit too risky. It seems that all the theoretical guarantees are also under the assumption that this initial choice is correct. Perhaps an alternative approach would be to retain a small collection of reference models (e.g., by performing individual goodness-of-fit tests for each and retain all models for which the null hypothesis was not rejected, while correcting for multiple testing)? - It seems that the proposed procedures could apply to both MMD and KSD and the theoretical guarantees could be derived in a fairly similar manner. In this case, could the proposed procedures also be adapted to other kernel-based tests, such as the ME and SCF tests of [6] and the FSSD test of [19] (especially since the relative goodness-of-fit tests of [17] were based on the ME and FSSD tests)? If so, the authors should provide a unified statement of the theoretical results that summarizes the requirements on the test statistic in order for the procedures to hold. If not, it would be helpful to point out the challenges involved. - In the legend of Figure 1, what does 'complete' and 'linear' indicate?

Reviewer 4



This paper proposes nonparameteric comparison tests for multiple (more than 2) models, and proves that these tests conrol either false positive rate or false discovery rate. The paper appears to be sound theoretically and has experiments which agree with the theoretical claims. There are some typos throughout the paper which sometimes affect the clarity, but they are not significant. However, I do have two major overarching concerns with the paper which I will detail below. Firstly, after reading this paper, I am not convinced that a statistical test for nonparametric relative multiple (more than 2) model comparison is something that I should be interested in as a machine learning researcher. Perhaps I should be interested in it, but I do not believe that the authors provide enough motivation to justify why we would need this type of test. If we have several models, isn't it good enough to perform a multiple goodness of fit test? It is even states that "the need for a reference model greatly complicates the formulation of the null hypothesis" - why bother with this new complication? I would be interested to hear the authors' opinion on this, and I would like to see this further motivated in the main text from the perspective of machine learning research. EDIT: Upon reviewing the authors' response, I am more convinced about the need for multiple model comparisons and the improvement that these tests provide over multiple GoF tests. I am confident that the authors will present this motivation in their paper, and thus I will raise my score. This leads me into my second point, which is that I am not sure NeurIPS is the appropriate venue for this type of paper. The submission appears too condensed to fit the NeurIPS format: notably, the algorithm describing how you *actually perform the statistical test* is shoved into the appendix, and the paper is lacking a conclusion. Furthermore, this type of relative multiple model comparison seems to be quite niche, and would possibly be more appropriate for a statistics journal. Another point that I was wondering - it is unclear what drives the improved results for the CelebA and Chicago datasets: is it really the multiple relative testing framework that has been introduced? Or is it the use of an improved metric (e.g. MMD vs. FID, or KSD vs. NLL)? Other miscellaneous points: Line 118 - I would change "one sample from R" to "samples from R" Line 168 - Why does the independence allow us to remove the constraint on Az? Line 271 - How is H_0: D(P_1, R) < D(P_2, R) true in the mean shift case? Wouldn't D(P_1, R) = D(P_2, R) here? Line 274 - What are the actual definition of P_1 and P_2 in this case? Supposedly they are MoGs that differ locally by rotation, but then what is the reference distribution R? Line 284 - Isn't it a bad idea to use a Gaussian kernel with the KSD? In goodness-of-fit testing, it was shown to be sometimes be unable to differentiate between two distributions; see (Gorham and Mackey, 2017). Small Typos Line 44 - Undefined acronyms (ME and SCF) Line 110 - Boldface x is introduced but doesn't appear anywhere else Line 126 - "Notation" -> "Notion" Line 264 - "RelPSI-MMD and RelPSI-MMD" doesn't make sense - I'm assuming one of them is supposed to be "RelPSI-KSD"? To summarize, I will comment on originality, quality, clarity, and significance: - *Originality*: The paper appears to be quite original. This work is placed in context with related work and the authors' contribution is clear. - *Quality*: The paper proposes a method, proves it to be correct in some sense, and then produces experiments which agree with its claims. I would consider that to be a submission of good quality. - *Clarity*: The paper is a bit dense and can be a bit hard to understand at times (likely because it was condensed to fit the NeurIPS format). However, the paper is very well organized (although lacking a conclusion) - *Significance*: As stated above, this is really where I think the submission is lacking. However, I am open to changing my mind if I can be convinced in the rebuttal. Reference: Gorham, Jackson, and Lester Mackey. "Measuring sample quality with kernels." Proceedings of the 34th International Conference on Machine Learning-Volume 70. JMLR. org, 2017.

[Author Response · NeurIPS 2019]

We thank all reviewers for their constructive comments. All typographic errors pointed out will be corrected accordingly.
Recall that $\mathcal{M} = \{P_i\}_{i=1}^l$ is the set of $l$ candidate models, $R$ is the unknown data generating distribution, $D$ is the
discrepancy measure (MMD or KSD), $J \in \arg\max_i D(P_i, R)$, and $\hat{J} \in \arg\max_i \hat{D}(P_i, R)$ (see L138 for details).

**Rev 1, 3, 7: Why compare more than two models?** Model comparison beyond two models is a more realistic scenario.
Given the availability of possible solutions, e.g. zoo of GANs, it is unlikely that a practitioner will only consider two
candidate models for a task. A popular approach is to rank candidate models by a fitness score (e.g., FID). These
estimated scores are correlated since they are computed on the same set of observations. Simply ranking these scores
without accounting for the randomness and correlation leads to uncontrolled false positive rate (FPR) e.g. Table 1
below, $B$ (true best) is not selected $1 - 83\% = 17\%$ of the time and **@[Rev 7]** for Experiment 3 (CelebA), Model 4
(the "best") is not selected $1 - 63\% = 37\%$ of the time. By contrast, the two proposed tests (RelPSI and RelMulti) have
a well controlled FPR and false detection rate (FDR), respectively, as noted by Rev 7. We will add to our manuscript.

**Rev 3, 7: Multiple goodness-of-fit testing vs multiple model comparison.** The two questions are fundamentally
different. In multiple goodness-of-fit testing, the goal is to determine whether $R$ (observed through samples) is in $\mathcal{M}$
i.e., find $P^* \in \mathcal{M}$ such that $D(P^*, R) = 0$. A PSI-based multiple goodness-of-fit test has been considered in [31] for
several candidate GAN models. Since all models are wrong (Box. 1976), it has led to the trivial result of the rejection
of all candidate models [31, section 5.3]. In multiple model comparison (our work), the goal is to find the model(s)
which has the lowest (not necessarily zero) discrepancy to $R$ i.e., find $P^* \in \arg\min_{P \in \mathcal{M}} D(P, R)$ with statistical
significance. While the former may be addressed by reducing it to $l$ individual goodness-of-fit tests (one for each
candidate), the latter problem is more complicated since *finding $P^*$ requires comparing $l$ correlated estimates of $D$*.

**All reviewers**: **Why not use the previous relative tests?** The relative model comparison tests RelMMD, RelKSD (for
$l = 2$) considered in [4, 17] require the practitioner to choose the ordering of models; that is, one has to decide a priori
$H_0: D(P_1, R) \leq D(P_2, R)$ or $H_0: D(P_2, R) \leq D(P_1, R)$. It is not obvious how one would use these relative tests to
find the best model(s) when $l > 2$. On the other hand, our proposed tests automatically determine the index $\hat{J}$ of the
best model, and take into account the fact that the data used to find $\hat{J}$ are the same as the data used for testing each
model against $P_{\hat{J}}$, creating the conditional null hypothesis (see L152, L169). This is the complication that did not exist
in the previous relative tests, and is the crux of our proposal. **@[Rev 1] L179-180**, **@[Rev 7] L168**: The conditional
$H_0$ reduces to the standard unconditional $H_0$ if the data used to find $\hat{J}$ are independent of the test data (i.e., conditioning
on an independent event $\boldsymbol{Az}$). The independence can be achieved by data splitting, which is the basis of the proposed
RelMulti.

**Rev 2: L1, positive and negative**. We follow the convention that when a test declares a significant result, it is positive.
Thus model $P_i$ is assigned positive when our test declares that it is worse than the best model (i.e., reject the null
hypothesis). **L95, Mild conditions:** If $\mathbb{E}_p[k(x, x)] < \infty$, then the mean embedding $\mu_p$ exists [26]. In particular,
if $k$ is bounded (e.g., IMQ kernel, Gaussian kernel), $\mu_p$ always exists. **L231, TPR and random sample:** TPR is
defined (in Appendix A) as the **population expectation** of the proportion of number of true positive models that
are declared as positive, and is not random. **L234, definition of $\mu$:** We define $\mu := D(P_1, R) - D(P_2, R)$. The
discrepancy measure $D$ can be MMD or KSD. **L271, $H_0$ is true:** In Experiment 1, $H_0 : D(P_1, R) \leq D(P_2, R)$ holds
since $D(P_1, R) = D(P_2, R)$ (there is a typo on L267). **L288, sampling variability:** For all our experiments, we
averaged the results over at least 100 trials (for Fig 1, it was 300 trials), with new samples redrawn in each trial.

**Rev 3: Selected reference is not the best**. It is true that the selection is noisy
and we can pick a worse model than the actual best, i.e. $\hat{J} \neq J$ (assuming
the best is unique). In this case, "$H_{0,i}^{\hat{J}} : D(P_{\hat{J}}, R) \geq D(P_i, R) \mid P_{\hat{J}}$ selected"
will hold for a larger portion of the tests, and will only result in lower TPR. In
particular, FPR is not affected. See Table 1. We emphasize that our **theoretical**
**results** do not make an assumption that the reference is correctly selected. It
is accounted for in TPR/FPR calculations and an incorrect rejection is made
with probability no larger than $\alpha$. **ME, FSSD, SCF:** We will provide a unified
statement in the revised version. "**Complete**" refers to the complete U-statistic
estimator and "**linear**" refers to the linear time estimator of [14, Section 6].

**Rev 7: Gaussian kernel with KSD**. [13] shows that if the KSD with a Gaus-
sian kernel is used to measure the discrepancy between a collection of $n$ points
$X_n$ from a non-convergent MCMC and a distribution $p$, then vanishing KSD
does not imply that $X_n \sim p$. This is an issue only when $X_n$ does not follow any
distribution at all. It is irrelevant for goodness-of-fit/model comparison testing
since $X_n$ is assumed to follow a proper distribution. The KSD goodness-of-fit
test will detect any discrepancy asymptotically (see Proposition 4.2 of [23]).

| $P_{\hat{j}} =$ | $A$ | $B$ | $C$ |
|---|---|---|---|
| Sel | 16% | 83% | 1% |
| CTPR | .115 | .271 | .009 |
| CFPR | .010 | 0 | 0 |
| $A$ | 0 | .065 | .017 |
| $B$ | .010 | 0 | 0 |
| $C$ | .229 | .477 | 0 |

Table 1: Results of the toy experiment of Rev 3 using RelPSI-MMD. Results averaged over 5000 trials with sample size 100. Sel is the proportion of times a particular $\hat{J}$ is selected. CTPR (and CFPR) is the empirical TPR (and FPR) conditioned on the selected $\hat{J}$. The bottom half shows rejection rates of each model for different $P_{\hat{j}}$. $\alpha = 0.05$. We estimate FPR = 0.001 and TPR = 0.2428.

[Meta-Review · NeurIPS 2019]

The reviewers agree that this submission represents an important contribution to the field. Please be sure to carefully review and address the concerns of all reviewers in the revision.